# Technical Note: two-component *Electrical Conductivity*-based hydrograph separa*T*ion employing an *EXP*onential mixing model (*EXPECT*) provides reliable high temporal resolution young water fraction estimates in three small Swiss catchments

Alessio Gentile[1], Jana von Freyberg[2,3], Davide Gisolo[1], Davide Canone[1], and Stefano Ferraris[1]

[1]Interuniversity Department of Regional and Urban Studies and Planning (DIST), Politecnico and Università degli Studi di Torino, 10125, Torino, Italy
[2]School of Architecture, Civil and Environmental Engineering, EPFL, 1015, Lausanne, Switzerland
[3]Mountain Hydrology and Mass Movements, Swiss Federal Institute for Forest, Snow and Landscape Research (WSL), 8903, Birmensdorf, Switzerland

*Correspondence to*: Alessio Gentile (alessio.gentile@polito.it )

**Abstract.** The young water fraction represents the portion of water molecules in a stream that have entered the catchment relatively recently, typically within 2-3 months. It can be reliably estimated in spatially heterogeneous and nonstationary catchments from the amplitude ratio of seasonal isotope ($\delta^{18}O$ or $\delta^{2}H$) cycles of streamwater and precipitation, respectively. Past studies found that young water fractions increase with discharge ($Q$), thus reflecting the higher direct runoff with wetter catchment conditions. The rate of increase in young water fraction with increasing $Q$, defined as the discharge sensitivity of the young water fraction ($S^{*}_d$), can be useful for describing and comparing catchments' hydrological behaviour. However, the existing method for estimating $S^{*}_d$, that only uses biweekly isotope data, can return highly uncertain and unreliable $S^{*}_d$ when the streamwater isotope data are sparse and don't capture the entire flow regime. Indeed, the information provided by isotope data depends on when the respective sample was taken. Accordingly, the low sampling frequency results in information gaps that could potentially be filled by using additional tracers sampled at higher temporal resolution.

By utilizing high-temporal-resolution and cost-effective electrical conductivity (EC) measurements, along with information obtainable from seasonal isotope cycles in streamwater and precipitation, we develop a new method that can estimate the young water fraction at the same resolution as EC and $Q$ measurements. These high-resolution estimates allow for improvements in the estimates of the $S^{*}_d$. Our so-called *EXPECT* method is built upon three key assumptions:

- A mixing relationship consisting in an exponential decay of streamwater EC with increasing young water fraction. It has been constructed based on the relationship between flow-specific young water fractions and EC.

- The two-component EC-based hydrograph separation technique, by using the above-mentioned exponential mixing model, can be used for a time-source partitioning of streamwater in young (transit times < 2-3 months) and old (transit times > 2-3 months) water.

- The EC value of the young water endmember ($EC_{yw}$) is lower than that of the old water endmember ($EC_{ow}$).

Selecting from measurements reliable values of $EC_{yw}$ and $EC_{ow}$ to perform this unconventional EC-based hydrograph separation is challenging, but the combination of information derived from the two tracers allows estimating the endmembers values. The two endmembers have been calibrated by constraining the unweighted and flow-weighted average young water fractions obtained with the EC-based hydrograph separation to be equal to the corresponding quantities derived from the seasonal isotope cycles.

We test the *EXPECT* method in three small experimental catchments in the Swiss Alptal valley by using two different temporal resolutions of $Q$ and EC data: sampling-resolution (i.e., we only consider $Q$ and EC measurements during dates of isotope sampling) and daily-resolution. The *EXPECT* method has provided reliable young water fraction estimates at both temporal resolutions, from which a more accurate discharge sensitivity of young water fraction ($S_d^{EXP}$) could be determined compared to the existing approach. Also, the method provided new information on $EC_{yw}$ and $EC_{ow}$, yielding calibrated values that fall outside the range of measured EC. This suggests that streamwater is always a mixture of young and old water also during very high or very low wetness conditions. The calibrated endmembers revealed a good agreement with both endmembers obtained from an independent method and EC measurements from groundwater wells.

For proper use of the *EXPECT* method, we have highlighted the limitations of EC as a tracer, identified certain catchment characteristics that may constrain the reliability of the current method and provided recommendation about its adaptation for future applications in other catchments than those investigated in this study.

## 1    Introduction

Environmental tracers in catchment studies are used for understanding the age, the origin, and pathways of water in natural environments (Kendall and McDonnell, 1998). Among tracers, hydrologists use the stable water isotopes ($^{18}$O and $^{2}$H) because they are constituent part of the water molecules and hence they are naturally present in precipitation (Kendall and McDonnell, 1998). The isotopic composition in precipitation ($C_P$) generally shows a pronounced seasonal cycle (Dansgaard, 1964). Catchment storage acts as a filter on this input seasonal cycle, so that the isotope cycle in streamwater ($C_S$) is damped and lagged compared to that in precipitation (McGuire and McDonnell, 2006). The delay and damping we observe in the streamwater cycle is caused by the advection and dispersion of stable water isotopes that reach the catchment with precipitation, thus reflecting the water mixing, diversity of flow paths and their velocities (Kirchner, 2016a; McGuire and McDonnell, 2006).

Kirchner (2016a, b) proposed a new water age metric directly related to the amplitudes ratio of the seasonal isotope cycles in streamwater and precipitation: the young water fraction, i.e., the portion of runoff younger than roughly 2-3 months. The precipitation isotope cycle amplitude ($A_P$) is generally estimated through a robust fit of a sine function on the isotopic composition of precipitation samples by using the precipitation amount associated to each sample as weight for reducing the influence of low-precipitation events (von Freyberg et al., 2018a; Kirchner, 2016a). The streamwater isotope cycle amplitude is estimated through a robust fit of a sine function on the isotopic composition of streamwater samples with

or without using discharges *(Q)* at the sampling times as weights (von Freyberg et al., 2018a). Please note that hereafter the symbol "*" indicates a streamflow-weighted variable. Therefore, it is necessary to distinguish between the unweighted and the flow-weighted streamwater amplitude ($A_S$ and $A{*}_S$, respectively; see supplementary material for further details) and, accordingly, between the unweighted and the flow-weighted young water fraction ($F_{yw}$ and $F{*}_{yw}$, respectively).

Recently, Gallart et al. (2020b) proposed a method for estimating the rate of increase in young water fraction with
increasing *Q* by fitting the sinusoid function, with amplitude $A{*}_S(Q) = A_P\, F{*}_{yw}(Q)$, directly to the isotopic data of stream water (Eq. 1):

$$C_S(Q, t) = A_S^*(Q)\sin(2\pi f t - \varphi_S^*) + k_S^* =$$
$$= A_P\big[F_{yw}^*(Q)\big]\sin(2\pi f t - \varphi_S^*) + k_S^* =$$
$$= A_P\big[1 - (1 - F_0^*)\exp(-Q\,S_d^*)\big]\sin(2\pi f t - \varphi_S^*) + k_S^* \tag{1}$$

Where $F_0^*$, $S_d^*$, $\varphi_S^*$ and $k_S^*$ parameters are obtained through non-linear fitting. The $S_d^*$ (d mm$^{-1}$) parameter is defined as the discharge sensitivity of young water fraction, $F_0^*$ (-) is the virtual young water fraction when $Q = 0$, $\varphi_S^*$ (rad) is the phase of the seasonal cycle, *f* is the frequency (equal to 1 y$^{-1}$ for a seasonal cycle) and $k_S^*$ (‰) is a constant representing the vertical offset of the seasonal cycle. Referring to the expression enclosed in square brackets in Eq. (1), the young water fraction is
assumed to vary with discharge following an exponential-type equation that converges toward 1 at the highest flows (see supplementary material for additional methodological details), but which does not converge toward 0 at the lowest flows, thus theoretically admitting $F_0^* < 0$. Because of this mathematical relationship between young water fraction and *Q*, young water fraction time series can in theory be calculated at the same temporal resolution as *Q*. However, the uncertainties of such time series can be substantial because the underlying isotope data, due to the low sampling frequency, are generally not
able to capture the entire range of flow regimes, especially the (very) high flow rates (Xia et al., 2023). This becomes evident in Figs. 1 and 3 of Gallart et al. (2020b) where standard errors of flow-specific $F_{yw}$ are largest during the highest flows. From these considerations emerges the need for a new method to reliably estimate the time series of young water fractions, and to better constrain the discharge sensitivity of young water fractions at very low and very high flow conditions.

Multi-year stable water isotope datasets are typically available at relatively low (e.g., biweekly, or monthly) temporal
resolutions because of high costs for sampling and laboratory analysis (Mosquera et al., 2018). For the same reasons, high resolution isotopic datasets are often limited to relatively short time-windows (Wang et al., 2019). However, the information provided by isotope data depends on when the respective sample was taken (Wang et al., 2019). Consequently, sampling at low temporal resolution results in information gaps that could potentially be filled by using additional tracers sampled at higher temporal resolution. As a tracer, electrical conductivity (EC), which is a bulk measure of the major ions in water
(Riazi et al., 2022), can be measured over extended periods at high temporal resolution, while costs for installation and maintenance remain low (Cano-Paoli et al., 2019; Mosquera et al., 2018). However, EC is not a conservative tracer (as stable water isotopes) because it is affected by geochemical reactions and dissolution of reactive solutes in streamwater (Cano-

Paoli et al., 2019; Benettin et al., 2022). Because of these characteristics, the tracers EC and stable water isotopes complement each other well, and thus can be jointly used to constrain model parametrizations and to inform transit time models (Cano-Paoli et al., 2019; Benettin et al., 2022).

A time-source separation is generally performed using isotope hydrograph separation, IHS (Klaus and McDonnell, 2013), while major ions (approximated by EC) have been previously used for geographic-source separation in endmember mixing analysis (Hooper, 2003; Penna et al., 2017). Major ions concentration in streamwater derives from mineral weathering. Weathering processes can be viewed as a series of geochemical reactions influenced by characteristics of fluid movement, such as the contact time between the flowing water and mineral surfaces (Benettin et al., 2015, 2017). Thus, the longer a water particle remains within the subsurface, the higher its solute concentration (and thus EC) will be once it will be released as streamflow (Benettin et al., 2017). Indeed, Mosquera et al. (2016), investigating the mean transit time (MTT) of water and its spatial variability in the wet Andean páramo, found that the mean electrical conductivity is an efficient predictor of mean transit time in this high-elevation tropical ecosystem. More recently, Riazi et al. (2022), modelling the EC variation using a travel time distribution approach, assumed that the salinity of water in catchment storages is a function of water age. Ognjen Bonacci and Tanja Roje-Bonacci (2023) used EC measurements of a karst spring to estimate the time that water spent in the karst aquifer. In addition, Kirchner (2016b) stated that the concentration of reactive chemical species, such as EC, can be used to construct mixing relationship with young water fraction, which provides information about the water age. Overall, these studies suggest that EC may provide useful information on water age (Riazi et al., 2022). Indeed, past studies used EC for time-source hydrograph separation (HS) in event and pre-event water with promising results that favourably compared with those obtained from conservative tracers (Riazi et al., 2022). For instance, Laudon and Slaymaker (1997), applied HS in two small nested alpine /subalpine catchments by using different tracers ($\delta^{18}$O, $\delta^2$H, EC and silica) overall returning comparable results. Cey et al. (1998), with the aim of quantifying groundwater discharge in a small agricultural watershed, separated the hydrograph in event and pre-event water (assumed to be groundwater) obtaining only slight different results utilizing $\delta^{18}$O and EC. Pellerin et al. (2008) performed HS on 19 low-to-moderate intensity rainfall events in a small urban catchment through the use of EC, silica and $\delta^2$H obtaining similar outcome regardless of the tracer used. In a similar environment, Meriano et al. (2011) revealed a high level of agreement between flow partitioning results during a midsummer event using HS via $\delta^{18}$O and EC as tracers. Camacho Suarez et al. (2015), to identify the mechanisms of runoff in a semi-arid catchment, applied HS by using both EC and $\delta^{18}$O highlighting no major disadvantages by using EC. More recently, Mosquera et al. (2018) used the *TraSPAN* model to simulate storm flow partitioning in a forested temperate catchment revealing similar portions of pre-event water regardless of the tracer ($\delta^{18}$O and EC) used. Cano-Paoli et al. (2019), by investigating the streamflow separation into event and pre-event components in an alpine catchment, obtained consistent results by using $\delta^{18}$O, $\delta^2$H and EC. Lazo et al. (2023) showed that, in a tropical alpine catchment, the use of EC returned similar results of event and pre-event water than those obtained with $\delta^{18}$O for a wide range of flow conditions reflected by the 37 monitored rainfall-runoff events. Overall, the findings of these studies suggest a quasi-conservative behaviour of EC under a wide range of hydrological and lithological conditions, also if its behaviour depends on specific characteristics (e.g.,

water partitioning between the surface and the subsurface, spatial distribution of minerals and subsurface properties, kinetics of rock dissolution, individual ions concentrations) of each watershed (Laudon and Slaymaker, 1997; Benettin et al., 2022; Lazo et al., 2023). Nevertheless, these studies have been limited to compare results obtained by applying the HS with
different tracers but did not integrate the information obtainable from stable water isotopes and EC to generate new insights into transit times, hydrological processes, and the links between water quality and water age variations (Benettin et al., 2022, 2017).

In this regard, we develop here a new multi-tracer method which combines biweekly stable water isotopes data ($\delta^{18}O$) with EC measurements. This study aims at both reducing the standard error of $S_d^*$ and estimating the young water fraction at
higher temporal resolution than two weeks, which will lead to new insights in the catchments under study.

## 2    Material and Methods

### 2.1    Study sites and data set

To test the applicability of our method (section 2.2), we use data from the Erlenbach (ERL), Lümpenenbach (LUE) and Vogelbach (VOG) catchments, located in the pre-alpine Alptal valley in central Switzerland. The geographical
framework of the three study sites is reported in Fig. 1.

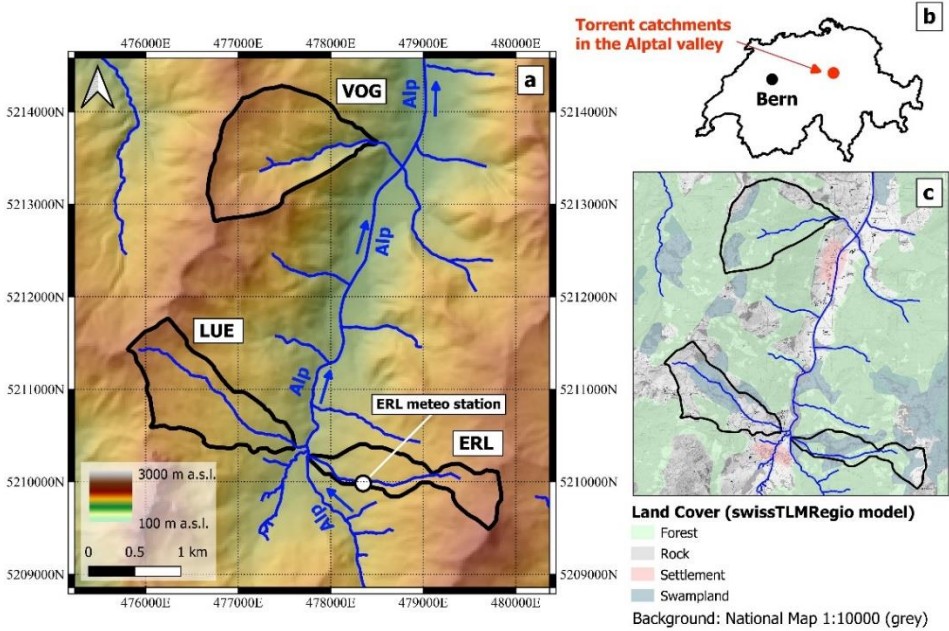

**Figure 1 a) Location of the three study catchments with indication of the stream networks and elevation (DHM25 ©swisstopo) as background. The Alp river is marked in the map with blue arrows indicating its flow direction. b) Location of the Alptal valley in Switzerland. c) Land cover of the three study catchments from the ©swissTLMRegio 2D landscape model.**

The three study catchments cover areas between 0.7 and 1.6 km$^2$, and mean elevation ranges from 1335 to 1359 m a.s.l (Table 1, Fig. 1a). Mean catchment slopes are 13.53°, 12.49° and 18.42° in the ERL, LUE and VOG catchments, respectively, but the hillslopes can be much steeper locally (20°-40°) (Stähli et al., 2021). According to the *swissTLMRegio* model (Fig. 1c), the ERL catchment is mainly constituted by forest (45%) and swampland (49%) which are the dominant classes also in the LUE (21% and 39%, respectively) and VOG (72% and 13%, respectively) catchments. Most of the

southern Alptal valley is characterized by shallow gleysols with low permeability that limit the deep infiltration of water and lead to shallow groundwater tables (Stähli et al., 2021). The percentage of soils with low storage capacity is about 4% in both ERL and LUE, while it is 51% in the VOG catchment; a large fraction of the soils is saturated ($\geq$ 95% in ERL and LUE, 49% in VOG; von Freyberg et al., 2018). The geological substratum of the three study sites consists mainly of sedimentary rock (flysch). The catchment area covered by Quaternary deposits is much higher in the ERL and LUE catchments than in

the VOG catchment (Table 1). Therefore, although the study catchments are located within close proximity, they differ in terms of soil wetness and unconsolidated sediments.

     The average hydro-climatic conditions are generally similar for all three catchments. The average annual precipitation in the period January 2000 - December 2015, based on interpolated data from the *PREVAH* model, was about 1853 mm, 1803 mm and 1800 mm at the ERL, LUE and VOG catchments, respectively (von Freyberg et al., 2018a). The

average monthly discharge is similar among the catchments: it is 138.9, 152.0, and 117.4 mm month$^{-1}$ at the ERL, LUE and VOG catchments, respectively (von Freyberg et al., 2018a). These watersheds reveal an hybrid hydro-climatic regime (Staudinger et al., 2017; von Freyberg et al., 2018a), since we observe an ephemeral snowpack formation (typically from December to April) that also during winter rapidly melts away so that the snowpack may not last throughout the entire winter season (Stähli et al., 2021).

Daily resolution $Q$ and streamwater EC data have been downloaded from the Swiss Federal Office for Forest, Snow and Landscape Research (WSL, Birmensdorf, Switzerland) data portal. We have estimated the $Q$-EC relationships with a log-type fit (Fig. 2). As daily $Q$ increases, daily EC decreases in the three study sites. This pattern arises due to the contribution of different sources (i.e., ages) of water to the stream. At the three study sites, stream discharge increases due to rainfall or snowmelt, which are generally low in EC, resulting in a dilution of streamwater EC. In addition, during wet

conditions (high $Q$), more rapid flow paths are activated leading to a prevalence of the younger hydrograph component. Because of the short interaction time with mineralized rocks and soils, young water can be assumed to be poor of dissolved ions (i.e., low EC). The other extreme, low $Q$ and high streamwater EC, occurs during baseflow conditions when the stream is mainly fed by old (i.e., highly-mineralized, high-EC) subsurface water (Schmidt et al., 2012).

     This study uses $F_{yw}$, $F^*_{yw}$, $F^Q_{yw}$ and $S^*_d$ (Table 2, Table 4), which were estimated in past studies (Gallart et al.,

2020b; von Freyberg et al., 2018a) by considering streamflow $\delta^{18}$O data from biweekly grab sampling over a period of approximately 5 years for the three study catchments. $F^Q_{yw}$ values refer to young water fractions estimated in discrete flow regimes (Kirchner, 2016b). Indeed, it is possible to separate the streamwater isotope data collected into different discharge ranges and fitting sinusoids separately to the isotope content in each range. For each of these individual flow regimes, this

method leads to obtaining the streamwater seasonal isotope cycles amplitude ($A^Q_S$) values that will be divided by $A_P$ to obtain

$F^Q_{yw}$ (von Freyberg et al., 2018a; Kirchner, 2016b). For more details about $F^Q_{yw}$ estimation, the reader is referred to Kirchner (2016b) and von Freyberg et al. (2018a).

**Table 1 Topographic, geological and hydro-climatic properties of the three study sites. Superscript "1" refers to data published in von Freyberg et al. (2018a);  Superscript "2" refers to data published in Gentile et al. (2023).**

| ID | ERL | LUE | VOG |
|---|---|---|---|
| [1]Area (km²) | 0.7 | 0.9 | 1.6 |
| [1]Mean elevation (range) (m a.s.l.) | 1359 (1117–1650) | 1336 (1092–1508) | 1335 (1038–1540) |
| [2]Mean slope (°) | 13.53 | 12.48 | 18.42 |
| [1]Saturated soils (%) | 0.95 | 0.96 | 0.49 |
| [2]Geological substratum | Sed. Rock (flysch) | Sed. Rock (flysch) | Sed. Rock (flysch) |
| [2]Areal fraction of Quaternary deposits (-) | 0.74 | 0.9 | 0.48 |
| [1]Regime (Staudinger et al., 2017) | hybrid | hybrid | hybrid |
| [1]Average precipitation (mm/month) | 162.4 | 157.1 | 162.2 |
| [1]Average discharge (mm/month) | 138.9 | 152 | 117.3 |
| [1]Period of isotope sampling | Jul 2010- May 2015 | Oct 2010-Nov 2015 | Jun 2010-Nov 2015 |

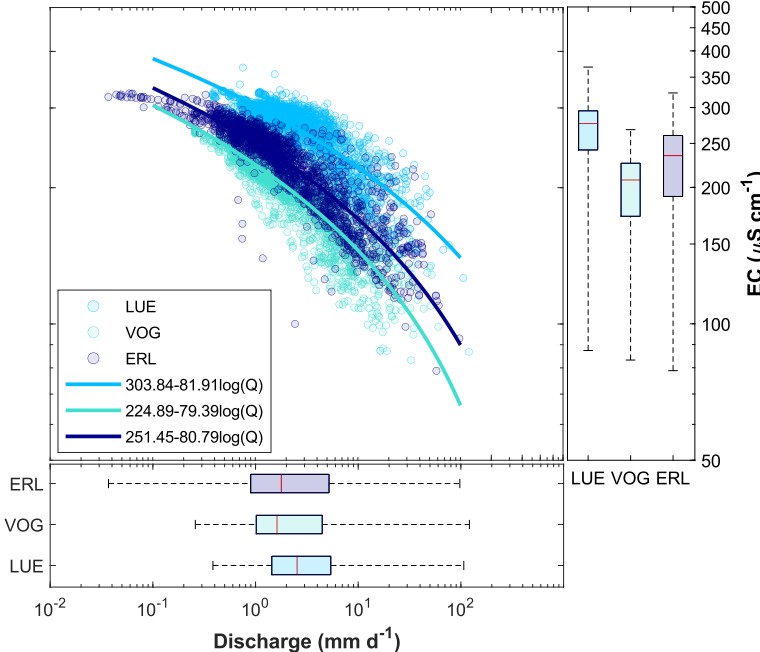


**Figure 2 Relation between daily EC and daily _Q_ for the three study sites. As discharge increases, the EC decreases in the three study catchments. This pattern arises mainly due to the age (source) of water contributing to the stream: if a substantial amount of recent, low-EC water contributes to streamflow during rainfall or snowmelt, streamwater EC decreases while discharge increases.**

**Table 2 Young water fractions of distinct flow regimes ($F^Q_{yw}$), as well as average unweighted and flow-weighted young water fractions ($F_{yw}$ and $F^*_{yw}$, respectively) with corresponding standard errors (SE). The number of samples used for estimating $F^Q_{yw}$ alongside the median $Q$ and EC of each flow regime are also reported. These data, excluding the median EC, were previously obtained by von Freyberg et al. (2018a).**

| Catch. ID | Q (range) | n° samples | Median $Q$ (mm d⁻¹) | Median EC (µS cm⁻¹) | $F^Q_{yw} \pm SE$ | $F_{yw} \pm SE$ | $F^*_{yw} \pm SE$ |
|---|---|---|---|---|---|---|---|
| ERL | Q (0-25%) | 35 | 0.42 | 274.68 | 0.294±0.039 | 0.37 ± 0.03 | 0.49 ± 0.03 |
| | Q (25-50%) | 35 | 0.93 | 248.71 | 0.353±0.032 | | |
| | Q (50-75%) | 35 | 2.21 | 213.28 | 0.449±0.049 | | |
| | Q (75-100%) | 35 | 7.23 | 163.21 | 0.467±0.048 | | |
| | Q (80%) | 28 | 8.20 | 157.18 | 0.446±0.061 | | |
| | Q (90%) | 14 | 19.21 | 148.51 | 0.52±0.083 | | |
| LUE | Q (0-25%) | 33 | 1.11 | 298.95 | 0.189±0.024 | 0.25 ± 0.02 | 0.33 ± 0.03 |
| | Q (25-50%) | 33 | 1.81 | 287.73 | 0.205±0.029 | | |
| | Q (50-75%) | 33 | 3.56 | 266.39 | 0.363±0.039 | | |
| | Q (75-100%) | 33 | 7.68 | 210.88 | 0.356±0.051 | | |
| | Q (80%) | 27 | 9.16 | 205.84 | 0.35±0.057 | | |
| | Q (90%) | 14 | 12.59 | 192.03 | 0.403±0.075 | | |
| VOG | Q (0-25%) | 35 | 0.73 | 234.97 | 0.163±0.02 | 0.21 ± 0.02 | 0.31 ± 0.02 |
| | Q (25-50%) | 35 | 1.11 | 217.55 | 0.168±0.024 | | |
| | Q (50-75%) | 34 | 2.22 | 193.28 | 0.267±0.034 | | |
| | Q (75-100%) | 35 | 7.80 | 148.08 | 0.316±0.039 | | |
| | Q (80%) | 28 | 8.65 | 142.19 | 0.325±0.044 | | |
| | Q (90%) | 14 | 12.13 | 133.02 | 0.36±0.051 | | |

## 2.2 The *EXPECT* method: two-component *E*lectrical *C*onductivity-based hydrograph separa*T*ion employing an *EXP*onential mixing model

The young water fraction may be useful in inferring chemical processes from streamflow concentrations of reactive chemical species (Kirchner, 2016b). Indeed, since it is known how the fraction of young water varies in discrete flow regimes, it is possible to construct mixing relationship between $F^Q_{yw}$ and the concentration of reactive chemical species (Kirchner, 2016b). Accordingly, we calculate the median EC within each individual discharge range, reported in Table 2, and we investigate how the median EC varies with $F^Q_{yw}$ (Fig. 3).

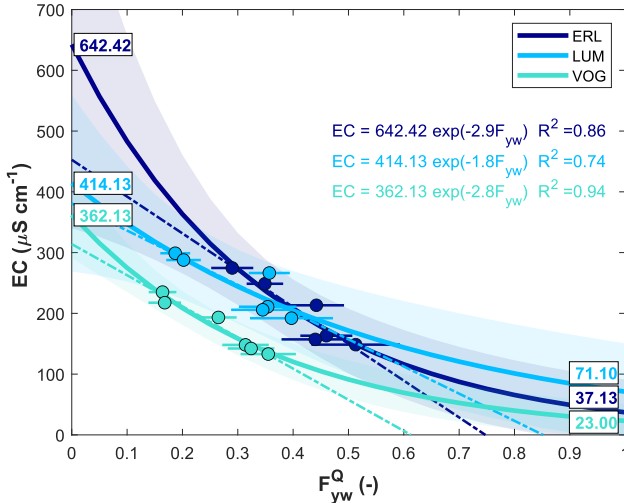

**Figure 3 Median flow-specific EC against $F^Q_{yw}$ for the three study catchments. Horizontal bars indicate the $F^Q_{yw}$ standard error. The solid lines indicate the exponential fits (which expressions with corresponding $R^2$ are also indicated). The coloured areas indicate the 90% prediction bounds. The textboxes corresponding to $F^Q_{yw}= 0$ indicate a first-order estimate of old water endmembers ($EC^{raw}_{ow}$) by using the exponential expression. Similarly, the textboxes corresponding to $F^Q_{yw}= 1$ indicate a first-order estimate of young water endmembers ($EC^{raw}_{yw}$) by using the exponential expression. The dashed lines indicate the linear fits to data that point to a negative EC endmember of young water (i.e., EC value corresponding to $F^Q_{yw}= 1$)**

As visible in Fig. 3, the relationship between $F^Q_{yw}$ and median flow-specific EC is well described by an exponential mixing model. Indeed, the widely used linear mixing model proves to be poorly suited here since it is pointing to a negative EC endmember of young water (i.e., EC value corresponding to $F^Q_{yw}= 1$, Fig. 3). This will be thoroughly discussed in the Appendix A. By considering the exponential mixing model, we can estimate the "idealized" old water and young water endmembers evaluating the fitted exponential expressions for $F^Q_{yw}= 0$ and $F^Q_{yw}= 1$, respectively. Accordingly, a first order estimate of the two endmembers ($EC^{raw}_{ow}, EC^{raw}_{yw}$, respectively) is reported in Fig. 3 and Table 3. It is evident that the measured $F^Q_{yw}$ for the three study catchments ranges from approximatively 0.1 to 0.5 (Fig. 3). Since the measurable range of young water fractions is not wide enough, the parameters estimated with the exponential fit are highly uncertain since the curve is poorly constrained at very low (< 0.1) and very high (> 0.5) young water fractions. In this regard, we propose hereafter a new methodology to estimate the EC endmembers of young and old water, respectively, and to perform a continuous hydrograph separation with an alternative mixing model.

The definition of the fraction of the streamflow younger than a threshold age (varying modestly from 2 to 3 months) at the generic time $t_i$, $F_{yw}(t_i)$, implicitly defines the existence of a complementary fraction of streamflow older than that threshold age at the same time $t_i$, $F_{ow}(t_i)$. Thus, mass conservation requires:

$$F_{yw}(t_i) + F_{ow}(t_i) = 1 , \tag{2}$$

To estimate $F_{yw}(t_i)$, and thus $F_{ow}(t_i)$, we use EC as a tracer to separate the hydrograph into young (transit time < 2-3 months) and old (transit time > 2-3 months) water. A solid support from the scientific literature justifying the use of EC for a time-source hydrograph separation has been illustrated in Section 1.

As suggested by the analysis reported in Fig. 3, to perform the hydrograph separation, we assume that streamwater EC at the generic time $t_i$, $EC(t_i)$, decreases exponentially with increasing $F_{yw}(t_i)$:

$$EC(t_i) = EC_{ow}e^{-aF_{yw}(t_i)} , \qquad (3)$$

where, $EC_{ow}$ is the old water EC endmember and $a$ is a parameter. The exponential decay proposed in Eq. (3) guarantees a realistic scenario for the case $F_{yw}(t_i) = 0$, i.e. streamflow contains only old water ($F_{ow}(t_i) = 1$) and streamwater EC is equal to $EC_{ow}$ ($EC(t_i) = EC_{ow}$). Conversely, if $F_{yw}(t_i)$ is equal to 1, streamflow is made up entirely of young water. Accordingly, we can include the following condition: if $F_{yw}(t_i) = 1$, $EC(t_i) = EC_{yw}$ where $EC_{yw}$ is the young water EC endmember (Eq. 4):

$$EC_{yw} = EC_{ow}e^{-a} , \qquad (4)$$

Furthermore, we assume $EC_{yw} < EC_{ow}$. A solid support from the scientific literature for this assumption will be illustrated in Section 3.1 alongside the discussion of the results.

By further considering the law of water mass conservation (Eq. 2), it is possible to solve the system of three equations (Eq. 2, 3, 4) with three variables ($a, F_{yw}(t_i), F_{ow}(t_i)$), thus obtaining the explicit expression of $a$ (Eq. 5) and, accordingly, of $F_{yw}(t_i)$ (Eq. 6).

$$a = -\ln\left(\frac{EC_{yw}}{EC_{ow}}\right), \qquad (5)$$

$$F_{yw}(t_i) = \frac{\ln\left(\frac{EC(t_i)}{EC_{ow}}\right)}{\ln\left(\frac{EC_{yw}}{EC_{ow}}\right)}, \qquad (6)$$

The main difficulty in applying Eq. (6) to estimate $F_{yw}(t_i)$ is that we generally cannot accurately determine the endmembers $EC_{yw}$ and $EC_{ow}$ neither from the analysis reported in Fig. 3 nor from measurements. Indeed, such endmembers correspond to the (rare) scenarios in which $F_{yw}(t_i)$ is either 0 or 1. The first scenario ($F_{yw}(t_i) = 0$) might occur only after prolonged periods without rainfall or snowmelt while the second scenario ($F_{yw}(t_i) = 1$) is unlikely to occur in most natural catchments where baseflow is usually older than 3 months (Gentile et al., 2023), and thus we cannot directly measure $EC_{yw}$ (Kirchner, 2016b). In this regard, we present hereafter a novel methodology to estimate the endmembers. Such methodology lays its foundations on the statement that the isotope-based $F_{yw}$ and $F^*_{yw}$, Eq. (7.1) and Eq. (7.2), see supplementary material for further details, accurately estimate the unweighted and the flow-weighted average young water fractions in streamflow, respectively (Kirchner, 2016b).

$$F_{yw} = \frac{A_S}{A_P} \tag{7.1}$$

$$F_{yw}^* = \frac{A_S^*}{A_P} \tag{7.2}$$

Accordingly, if we know the young water fraction over a generic time step $t_i$, $F_{yw}(t_i)$ (e.g., daily young water fraction), we can calculate the unweighted and the flow-weighted average young water fraction in streamflow through Eq.
(8.1) and Eq. (8.2), respectively:

$$\tilde{F}_{yw} = \frac{\sum_{i=1}^{n} F_{yw}(t_i)}{n} \simeq F_{yw} \tag{8.1}$$

$$\tilde{F}_{yw}^* = \frac{\sum_{i=1}^{n} Q(t_i)F_{yw}(t_i)}{\sum_{i=1}^{n} Q(t_i)} \simeq F_{yw}^* \tag{8.2}$$

where $n$ is the number of time-steps (e.g., days) in the period of isotope sampling and $Q(t_i)$ is the discharge at the time $t_i$ (e.g., daily discharge). The hat "~" symbol is simply used to visually differentiate the average young water fractions
obtained with both approaches. Please, note that Eq. (8.2) was previously presented in Gentile et al. (2023).

We therefore determine $EC_{yw}$ and $EC_{ow}$ through calibration, respecting the following three constraints:

i. $EC_{ow}$ and $EC_{yw}$ are greater than or equal to 0.

ii. $\tilde{F}_{yw}$, where $F_{yw}(t_i)$ is obtained through Eq. (6), must match the $F_{yw}$ estimated with the amplitude ratio technique (Eq. 7.1).

iii. $\tilde{F}_{yw}^*$, where $F_{yw}(t_i)$ is obtained through Eq. (6), must match the $F^*_{yw}$ estimated with the amplitude ratio technique (Eq. 7.2).

In summary, we perform a constrained EC-based hydrograph separation in which the two endmembers ($EC_{yw}$ and $EC_{ow}$) are calibrated through an optimization procedure. Specifically, we use the © Matlab *fmincon* solver, the *sqp* (sequential quadratic programming) algorithm, within the *GlobalSearch* procedure that runs repeatedly the local solver for
generating a global solution. To satisfy point i), we search the endmember values within the range $[0, +\infty)$. We consider $\infty$ as upper limit since catchments can also have immobile storages that potentially will never participate to the water cycle (Staudinger et al., 2017). In addition, we calibrate the EC endmembers by minimizing the following objective function, which is designed for satisfying points ii) and iii).

$$obj = \frac{\left(\tilde{F}_{yw} - F_{yw}\right)^2 + \frac{F_{yw}^*}{F_{yw}}\left(\tilde{F}_{yw}^* - F_{yw}^*\right)^2}{\left(1 + \frac{F_{yw}^*}{F_{yw}}\right)}, \tag{9}$$

We are giving a greater weight to the second term, $\left(\tilde{F}_{yw}^* - F_{yw}^*\right)^2$. The weight is proportional to how much $F^*_{yw}$ is higher than $F_{yw}$, since Gallart et al. (2020a) showed that the flow-weighted analysis produces a less biased estimation of young water fraction. The outputs of the optimization procedure are the calibrated young water and old water endmembers ($EC_{yw}^{opt}$ and $EC_{ow}^{opt}$, respectively). Subsequently, we calculate the $F_{yw}^{opt}$ (at every time step $t_i$) with Eq. (6) by using the optimal endmembers ($EC_{yw}^{opt}, EC_{ow}^{opt}$) and we plot $F_{yw}^{opt}$ against $Q$, thus visualizing an empirical relationship between the two

variables. Finally, we fit the expression enclosed in square brackets in Eq. (1) (corresponding to Eq. (6) from Gallart et al. (2020b)) to our $F_{yw}^{opt}$ data:

$$F_{yw}^{opt} = 1 - (1 - F_0^{EXP}) \exp(-Q\, S_d^{EXP}),$$    (10)

We then compare the discharge sensitivity, $S_d^*$, previously determined using only streamwater isotope data (see Eq. 1), and the discharge sensitivity, $S_d^{EXP}$, determined from Eq. (10). We further compare our results to the $F^Q_{yw}$ values (Table

2) previously obtained by von Freyberg et al. (2018a).

We apply our method at two different time-resolutions that are reflected in our data set: daily resolution (DR) and sampling resolution (SR). At DR, $EC(t_i)$ and $Q(t_i)$ refer to daily average EC and $Q$, respectively, and thus, $F_{yw}(t_i)$ is the average young water fraction of each day. At SR, please note that the "EC samples" are not referring to physical samples in this specific application. Accordingly, $EC(t_i)$ and $Q(t_i)$ are obtained by sub-setting those EC and $Q$ values from the daily

time series that correspond to the time of isotope sampling. In this sense, we can say that the number of EC samples and isotope samples is the same. Nevertheless, the method can be potentially applied at SR in catchments in which EC is only measured from water samples. At SR, $F_{yw}(t_i)$ values are estimated only for those days on which an isotope sample was taken.

We quantify the uncertainty of $EC_{yw}^{opt}$ and $EC_{ow}^{opt}$ by repeating the global optimization procedure by sampling

randomly 10000 couples of $F_{yw}$ and $F^*_{yw}$ from the intervals $F_{yw} \pm$ SE and $F^*_{yw} \pm$ SE, respectively. The SE values are reported in Table 2. The random sampling assumes that the values within the two intervals have a Gaussian probability of extraction, thus favoring the sampling of the core values. Therefore, we obtain 10000 couples of endmembers of which we compute statistics. We further calculate the uncertainty of $F_{yw}^{opt}(t_i)$: we apply Eq. (6) using the 10000 couples of endmembers, thus obtaining 10000 $F_{yw}^{opt}(t_i)$ values at each time step $t_i$, of which we calculate the standard deviation.

Please, note that the initial conceptualization of the mixing model was based on testing the hydrograph separation by using the 2-component endmember linear mixing approach with EC as tracer (e.g., Cano-Paoli et al. 2019). As could already be inferred from Fig. 3, this approach was not successful because it can represent only a limited hydrological behaviour of catchments that does not capture that of our three study catchments. A detailed explanation of the limits regarding the linear mixing model is provided in the appendix A of this paper.

Last, but not least, since our method consists in a two-component *E*lectrical *C*onductivity-based hydrograph separa*T*ion employing an *EXP*onential mixing model, we decide to name it *EXPECT*. A schematic representation of the *EXPECT* method is reported in Fig. 4.

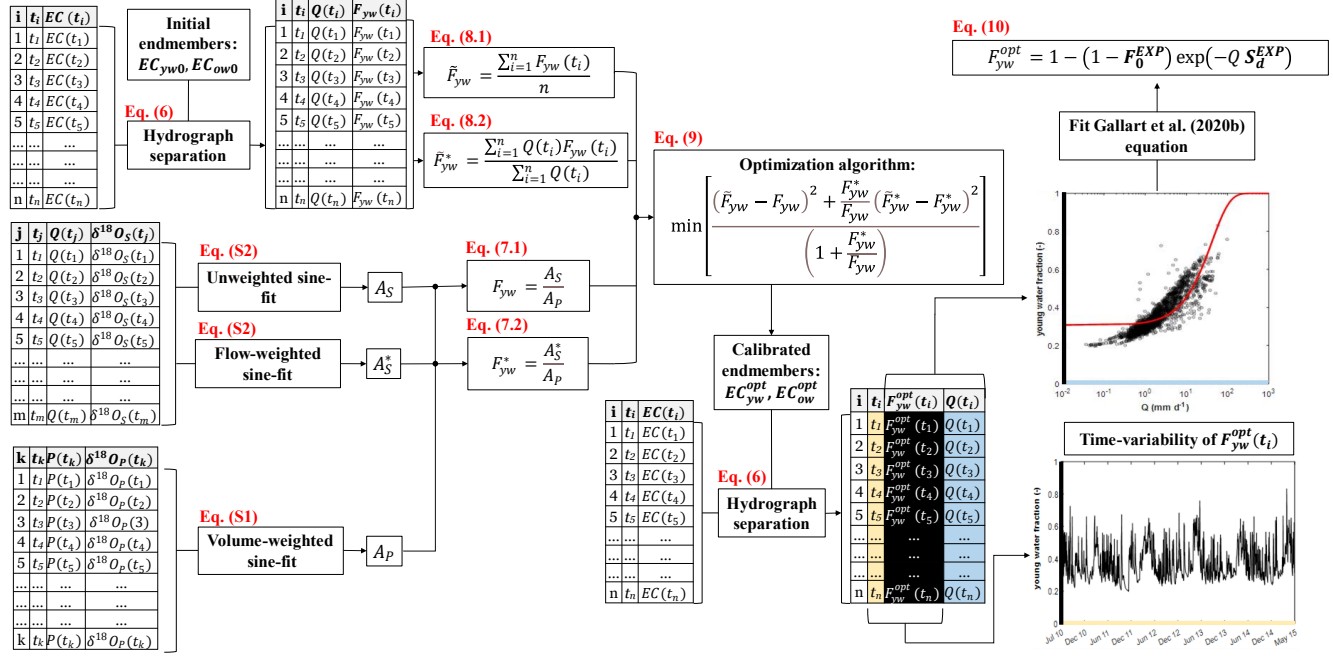

**Figure 4** Schematic representation of the *EXPECT* method. The subscript "P" refers to precipitation, while the subscript "S"
refers to streamwater. $P(t_k)$ indicates the volume of precipitation used for the volume-weighted fit of precipitation isotopes
($\delta^{18}O_p(t_k)$). The sampling times of $EC(t_i)$, $Q(t_i)$, $\delta^{18}O_S(t_j)$, $\delta^{18}O_P(t_k)$ may not be aligned, and consequently, the time series
typically have different lengths. Thus, the times $t_i$, $t_j$ and $t_k$ have different indices and usually n ≠ m ≠ k.

## 3    Results and Discussion

### 3.1    Physical likelihood of calibrated endmembers and discharge sensitivity of young water fraction

The application of the *EXPECT* method showed, at both daily and sampling resolution, that the old water EC
endmembers, $EC_{ow}^{opt}$, are about one order of magnitude larger than the young water EC endmembers, $EC_{yw}^{opt}$, for all three
experimental catchments (Table 3, Fig. 5). This result can be explained by considering that old water had longer contact with
mineral surfaces in the subsurface (Benettin et al., 2015, 2017), and thus weathering-derived solute concentrations (and
correspondingly EC) will be higher in old water compared to that in young water. Moreover, young and old streamwater
components can derive from different reservoirs in a catchment (Riazi et al., 2022). Among these reservoirs, old water is
generally assumed to represent groundwater. This is also supported by the fact that the fraction of baseflow (representing
groundwater contribution to streamflow) resulted to be complementary to young water fraction in the framework (including
the three Swiss catchments of this study) investigated by Gentile et al. (2023). In this regard, different papers that
characterized groundwater EC showed notable differences with EC of precipitation and/or meltwater. Indeed, Zuecco et al.
(2018), by investigating the hydrological processes in an alpine catchment, found that EC of rain water and of recent snow is

19.2 µS/cm and 12.2 µS/cm, respectively. Conversely, they found that groundwater from springs had an EC of 166 µS/cm. Moreover, by investigating the conceptualization of meltwater dynamics in an alpine catchment through hydrograph separation, Penna et al. (2017) defined the snowmelt endmember ranging from 2.9 to 15.3 µS/cm, the glacier melt endmember ranging from 2 to 2.7 µS/cm and the groundwater endmember ranging from 210 to 317.7 µS/cm (average values from springs or streams in fall/winter). These examples are intended to show that groundwater (main source of old water) generally reveals an EC value much higher (around 10÷100-fold) than other sources in a catchment that should preferentially contribute to the young streamwater component. Differences in young and old water EC endmembers can also be partially justified by looking at differences in event and pre-event water EC endmembers. Indeed, old (transit times > 2-3 months) water is a large fraction of pre-event (transit times > few days) water, whereas event water (transit times < few days) is a portion of young water (transit times < 2-3 months). Due to this overlap, it would not be surprising a similarity of the old water and pre-event water EC endmembers, as well as the young water and event water EC endmembers. Cano-Paoli et al. (2019) used streamwater EC to investigate hydrological processes in alpine headwaters by separating the hydrograph into event and pre-event water. In this regard, they defined the event water end-member equal to 8 µS/cm (Penna et al., 2014) and the pre-event water endmember equal to 95 µS/cm (mean value during baseflow conditions). Laudon and Slaymaker (1997), by investigating the hydrograph separation using EC at the lower station of an alpine catchment, defined the rain water EC endmember equal to 6.15 µS/cm and the pre-event water endmember equal to 39 µS/cm. However, young and old water EC endmembers are expected to be higher than event and pre-event water EC endmembers, respectively. Accordingly, these past results taken from the scientific literature support our assumption that $EC_{yw} < EC_{ow}$.

The highest $EC_{ow}^{opt}$ values were obtained for ERL (501 $\mu S \ cm^{-1}$, DR), and the lowest values in VOG (319 $\mu S \ cm^{-1}$, DR). The $EC_{ow}^{opt}$ values are in line with those measured in groundwater (Fig. 5): in a 6.8-m deep monitoring well at the ERL meteorological station, groundwater EC varies generally between 400 (spring-summer) and 500 $\mu S \ cm^{-1}$ (fall-winter; data not shown), whereas in a neighbouring catchment of ERL, EC in groundwater in up to 1.5 m depth was generally around 400-450 $\mu S \ cm^{-1}$ during no-snowmelt conditions (Kiewiet et al., 2020). The optimal endmembers are also in line with the first-order estimates of endmembers, $EC_{ow}^{raw}, EC_{yw}^{raw}$, derived from the exponential model fitted on median EC vs $F_{yw}^{Q}$ (Table 3, Fig. 3, Fig. 5), except for $EC_{ow}$ in ERL catchment. This can be explained by considering the high standard error (Table 3) of the parameter (corresponding to $EC_{ow}^{raw}$) in the exponential model that is for ERL (more than LUE and VOG) poorly constrained at low young water fractions (Fig. 3). The optimized EC values of the young water fractions appear slightly elevated compared to data derived from Central European Precipitation (Monteith et al., 2023). However, it is plausible to posit that the young water fraction encompasses some soil water with higher EC.

Fig. 5 shows further that the interquartile ranges (IQR) of the $EC_{ow}^{opt}$ empirical distributions are much larger than those of $EC_{yw}^{opt}$. Assuming that the solute concentration in streamwater increases with water age (Riazi et al., 2022), this can possibly be explained with the much wider range of transit times (from approximately 0.2 to ∞ y) of the old water compared

to that of young water (0 to 0.2 y). Consequently, the concentrations of weathering-derived solutes in old water are not only higher but also more variable than in young water.

Our method estimates the EC endmember values for the cases $F_{yw}(t_i) = 1$ and $F_{yw}(t_i) = 0$ that are generally difficult to determine experimentally, thus providing additional information about young and old water in the systems under study. In this regard, in each one of the three study sites, the theoretical endmembers $EC_{yw}^{opt}$ are lower than the minimum EC value measured in the streams; analogously, the calibrated $EC_{ow}^{opt}$ values are higher than the maximum measured EC value (boxplots *versus* horizontal dashed lines in Fig. 5). This is expected for a natural, heterogeneous system where incoming precipitation mixes with stored water, and thus streamwater never contains 100% young or old water, respectively. Instead, streamwater is a mixture of these two components. This is supported by the fact that $F^Q_{yw}$ cover only a limited range of young water fractions (roughly from 0.1 to 0.5). This result demonstrates that the choice of the old water endmember based on tracer values sampled during baseflow conditions can result in an underestimation of the theoretical old water endmember. Although these stream conditions suggest the prevalence of old water, if the percentage of old water is less than 100%, then the measured tracers still reflect some mixing (albeit limited) with young water.

**Table 3 Optimized endmembers obtained through the *EXPECT* method. 1st, 2nd, 3rd quartile (q1, q2 and q3, respectively) and IQR of optimized endmembers empirical distribution are also reported. First order estimates of endmembers derived from the exponential model fitted on median EC vs $F^Q_{yw}$ (see also Fig. 3) with related standard errors are reported on the right side of this table. Values are in $\mu S\ cm^{-1}$.**

| Time-resolution | ID | $EC_{yw}^{opt}$ | q1 | q2 | q3 | IQR | $EC_{ow}^{opt}$ | q1 | q2 | q3 | IQR | ID | $EC_{yw}^{raw} \pm SE$ | $EC_{ow}^{raw} \pm SE$ |
|---|---|---|---|---|---|---|---|---|---|---|---|---|---|---|
| Daily (DR) | ERL | 54.25 | 44.28 | 54.05 | 63.17 | 18.89 | 501.03 | 446.52 | 502.47 | 583.37 | 136.85 | ERL | 37.13±62.53 | 642.42±140.13 |
| | LUE | 51.08 | 37.27 | 50.67 | 65.02 | 27.75 | 449.79 | 411.12 | 450.29 | 504.31 | 93.19 | | | |
| | VOG | 29.71 | 23.79 | 29.45 | 35.13 | 11.34 | 318.82 | 300.33 | 319.92 | 345.73 | 45.4 | LUE | 71.10±139.60 | 414.13±62.81 |
| Sampling (SR) | ERL | 44.78 | 35.88 | 44.74 | 53.4 | 17.52 | 565.89 | 495.15 | 566.39 | 668.09 | 172.94 | | | |
| | LUE | 65.68 | 49.29 | 65.18 | 80.93 | 31.64 | 410.43 | 379.38 | 410.69 | 454.26 | 74.88 | VOG | 23.00±47.75 | 362.13±31.13 |
| | VOG | 32.25 | 25.64 | 31.41 | 37.27 | 11.63 | 315.23 | 299.56 | 318.53 | 342.67 | 43.11 | | | |

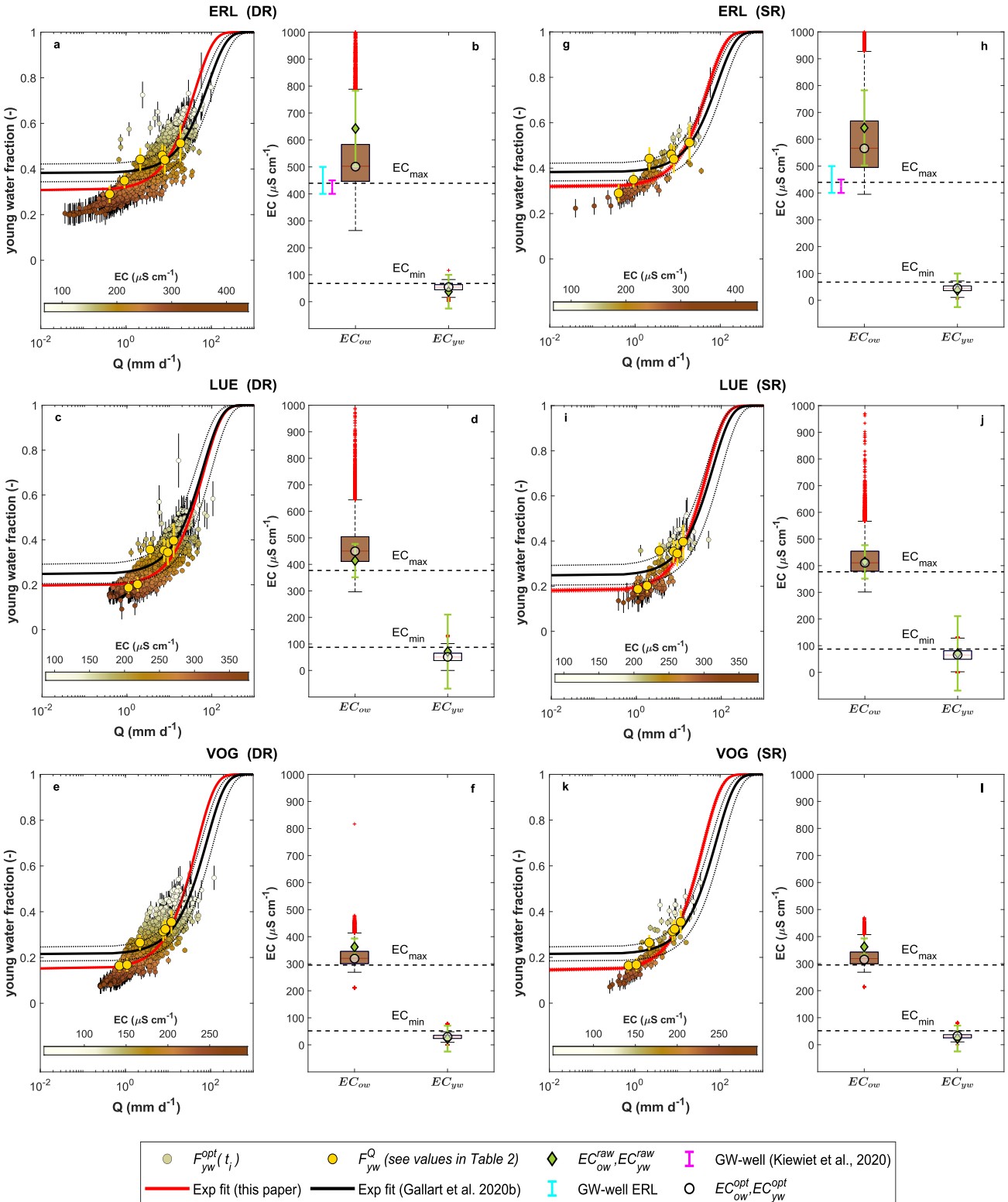

**Figure 5** $F_{yw}^{opt}(t_i)$- $Q(t_i)$ relation for the ERL, LUE and VOG study catchments at daily resolution (DR, panels a, c, e) and sampling resolution (SR, panels g, i, k), as well as the corresponding EC endmembers (b, d, f and h, j, l, respectively). The white-brown colour of the $F_{yw}^{opt}(t_i)$ points indicates the $EC(t_i)$ value. For comparison, average $F_{yw}^Q$ values of specific flow ranges (Table 2) are shown in yellow with related standard errors (yellow bars). The black curve represents the exponential-type fit by using parameters $S_d^*$ and $F_0^*$ previously obtained through non-linear fitting of Eq. (1) to streamwater isotope data by Gallart et al. (2020b). The red curve represents the exponential-type fit by using parameters $S_d^{EXP}$ and $F_0^{EXP}$ obtained in this study through non-linear fitting of Eq. (10) to $F_{yw}^{opt}$ vs $Q$. Black and red dashed lines indicate ±1 standard error. Panels b), d), f), h), j), l) show the boxplots of $EC_{yw}^{opt}$ and $EC_{ow}^{opt}$ derived from the endmember uncertainty analysis. The white dots indicate the optimal endmembers (obtained constraining the EC-based hydrograph separation using $F_{yw}$ and $F^*_{yw}$) used to calculate $F_{yw}^{opt}(t_i)$ through Eq. (6). $EC_{ow}^{raw}, EC_{yw}^{raw}$ (with related standard errors: green bars) have been superimposed for validation purposes along with measured EC range in two groundwater wells within (solid line in cyan) and nearby (solid line in magenta) the ERL catchment. The dashed black lines, labelled with $EC_{max}$ and $EC_{min}$, refer to the maximum and minimum EC values measured in the stream.

The estimated discharge sensitivity of the young water fraction, $S_d^{EXP}$, based on the *EXPECT* method satisfactorily describes the $F_{yw}^{opt}(t_i)$- $Q(t_i)$ relationships of the three catchments, as reflected by $R^2$ values of 0.58 and higher (Table 4; red curves in Figure 4). Moreover, the red curve also fits well the $F_{yw}^Q$ values of the distinct flow regimes (Table 2). By taking advantage of the consecutive $F_{yw}^{opt}(t_i)$ values at daily or sampling resolution, we better constrain the parameters of Eq. (10) at very low and very high discharges compared to the fit obtained with Eq. (1) that is using only streamwater $\delta^{18}$O data at sampling resolution (black curve in Fig. 5, Table 4; see the supplement for methodological details). As a result, our estimated discharge sensitivity $S_d^{EXP}$ is higher for the ERL and VOG catchments and similar (within error) for the LUE catchment compared to $S_d^*$, whereas our estimates of $F_0^{EXP}$ for all three sites are slightly smaller than the respective $F_0^*$ values obtained with Eq. (1).

We also find that the $S_d^{EXP}$ values obtained at SR can differ from those at DR. For LUE, $S_d^{EXP}$ at SR is larger than at DR (Table 4), whereas it is the other way around for ERL. Such differences can be attributed to the different flow regimes represented by the isotope samples that influences the EC endmember estimations at each site (Table 3). Moreover, at DR we are calibrating the EC endmembers by using $F_{yw}$ and $F^*_{yw}$ based on isotope data at SR. To be fully consistent in terms of temporal resolution, we theoretically need daily streamwater isotope data to derive $F_{yw}$ and $F^*_{yw}$. The influence of sampling frequency is one of the limitations of the *EXPECT* method as explained in section 3.3. Nevertheless, the $F_0^{EXP}$ values are consistent between the two temporal resolutions.

As can be seen in Fig. 5, the $F_{yw}^{opt}(t_i)$ values obtained with the EXPECT method form a data cloud around the idealized discharge sensitivity function of Eq. (10). Specifically, for a given discharge value, we obtain various $F_{yw}^{opt}(t_i)$ values, which can be explained by the delayed response of old water during precipitation events: while the young water

fraction is generally highest during the rising limb of the hydrograph, it decreases during the falling limb when old water reaches the stream (von Freyberg et al., 2018b)

**Table 4 Comparison of discharge sensitivity parameters obtained with the *EXPECT* method ($S_d^{EXP}$, $F_0^{EXP}$), by fitting Eq. (10) on $F_{yw}^{opt}$ data (the goodness of fit is indicated by $R^2$), and parameters obtained with the Gallart et al. (2020b) method ($S_d^*$, $F_0^*$) by fitting**
**Eq. (1) directly to the seasonal variation of the isotopic signal of stream water.**

| Time-resolution | Catch. ID | $F_0^* \pm SE$ (-) | $F_0^{EXP} \pm SE$ (-) | $S_d^* \pm SE$ (d mm$^{-1}$) | $S_d^{EXP} \pm SE$ (d mm$^{-1}$) | $R^2$ |
|---|---|---|---|---|---|---|
| | | Eq. (1), (Gallart et al., 2020b) | Eq. (10), this study | Eq. (1), (Gallart et al., 2020b) | Eq. (10), this study | this study |
| Daily (DR) | ERL | - | 0.3047±0.002 | - | 0.024±0.0005 | 0.62 |
| | LUE | - | 0.1948±0.0016 | - | 0.0155±0.0003 | 0.61 |
| | VOG | - | 0.1488±0.0016 | - | 0.0211±0.0004 | 0.64 |
| Sampling (SR) | ERL | 0.382±0.0387 | 0.317±0.0062 | 0.012±0.0034 | 0.0198±0.0016 | 0.64 |
| | LUE | 0.246±0.0429 | 0.1773±0.0073 | 0.016±0.0056 | 0.0223±0.0017 | 0.58 |
| | VOG | 0.214±0.03 | 0.1415±0.0056 | 0.012±0.0036 | 0.0252±0.0015 | 0.70 |

## 3.2 An immediate application of the *EXPECT* method: flow duration curves of young/old water and the temporal variability of young water fractions.

Because the *EXPECT* method allows for estimating young water fractions $F_{yw}^{opt}(t_i)$ at up to daily resolution, we can
determine the flow duration curves of young and old water discharge, respectively. Moreover, we calculate $Q_{50/50}$, i.e., the median discharge value at which $50 \pm 1\%$ of both young and old water exist in streamflow. In the ERL catchment, Fig. 6a shows that a shift from old-water dominated towards young-water dominated streamflow occurs for discharges larger than approximately 7.7 mm d$^{-1}$ ($Q_{50/50}$; Fig. 6a). In the LUE and VOG catchments, the streamflow contains more old water than young water for most of the flow regime (Fig. 6b, Fig. 6c); only for relatively few occasions, when $Q$ exceeds $Q_{50/50}$ (23.2
and 17.5 mm d$^{-1}$, respectively), the relative contribution of young water was slightly larger than that of old water.

By comparing $Q_{50/50}$ with the median stream discharge ($Q_{med}$), we observe that in all three study catchments $Q_{50/50}$ is higher than $Q_{med}$ (Fig. 6). This result suggests that more than 50% of the time a major proportion of old water reaches the stream. In both the LUE and VOG catchments, $Q_{50/50}$ is higher than in the ERL catchment, revealing that the LUE and VOG streams are longer dominated by old water than the ERL stream. This explains why the isotope-based average young water
fraction is higher in the ERL than in the LUE and VOG catchments (Table 2).

With the *EXPECT* method, the time-variability of $F_{yw}^{opt}(t_i)$ can be explored in detail, e.g. through comparing time series of $F_{yw}^{opt}(t_i)$ with those of other hydro-climatic variables (Fig. 7). Accordingly, we show hereafter a comparison between $F_{yw}^{opt}(t_i)$ and hydro-climatic observations at daily resolution of the ERL catchment since it has the most complete

hydro-climatic data set (including discharge, precipitation, snow depth and temperature measurements; all data available

from WSL) compared to the other two catchments. As visible from Fig. 7, daily young water fractions in the ERL catchment respond directly to precipitation events, which is further reflected by a strong positive correlation between $F_{yw}^{opt}(t_i)$ and the daily precipitation volumes ($\rho_{Spearman}$ = 0.41, p-value << 0.01 considering only days with precipitation, Fig. 8). We estimate that after a rainfall- or snowmelt event, the growth rate of $F_{yw}^{opt}(t_i)$ is on average 0.062±0.058 d$^{-1}$ (to reach the local $F_{yw}^{opt}(t_i)$ maximum next to the previous $F_{yw}^{opt}(t_i)$ local minimum, Fig. S1). On the other side, during the recession phase, the average

rate of decrease of $F_{yw}^{opt}(t_i)$ is -0.041±0.036 d$^{-1}$ (to reach the $F_{yw}^{opt}(t_i)$ local minimum next to the previous $F_{yw}^{opt}(t_i)$ local maximum, Fig. S1). Accordingly, $F_{yw}^{opt}(t_i)$ rapidly increases after an event (peak $F_{yw}^{opt}(t_i)$ is reached on average after 1.98±1.25 days), while it recedes slower during no-input days (the next minimum $F_{yw}^{opt}(t_i)$ is reached on average after 3.36±3.10 days). The largest daily young water fractions in the ERL catchment occurred during spring snow melt (March-May), suggesting that the melt water of the ephemeral snowpack is an important source of young water (since no relevant

water aging is observed in such snowpack) that flows off quickly in the stream (Gentile et al., 2023). Rapid surface runoff of snow melt can occur due to soil freezing (temperatures < 0°C) or high soil moisture contents (temperatures > 0°) both of which can limit infiltration (Harrison et al., 2021; Keller et al., 2017; Fig. 7). During the periods of snow accumulation and persistent snow cover, typically from November to February, $F_{yw}^{opt}(t_i)$ values were often as low as 0.3 and did not vary much (except during snowmelt and rain-on-snow events). Thus, streamflow in ERL was mainly composed of old water during this

period, likely originating from the soil- and groundwater storages.

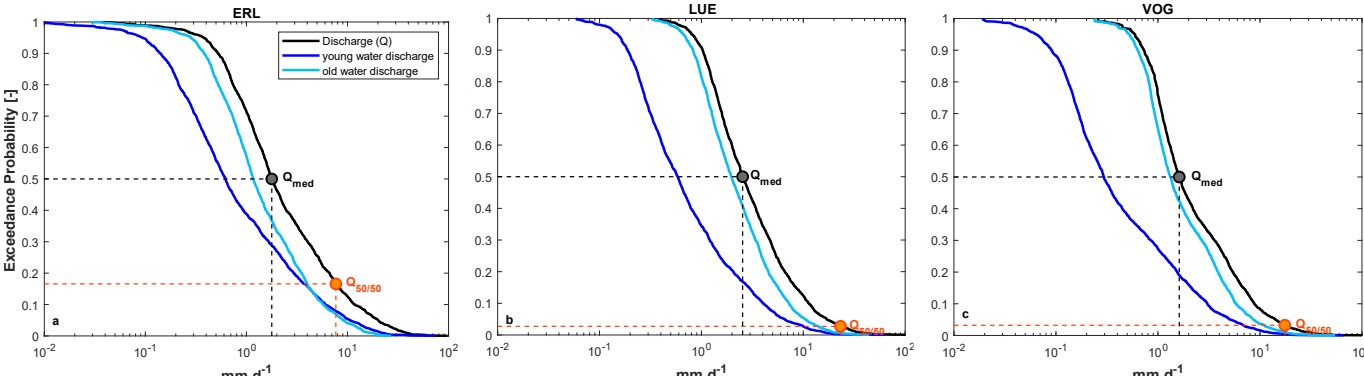

**Figure 6 Total flow, young flow and old flow duration curves of a) ERL, b) LUE and c) VOG catchments. Q$_{50/50}$ indicates the median discharge value at which 50 ± 1% of both young and old water exist in streamflow. *Q$_{med}$* represents the median stream discharge.**

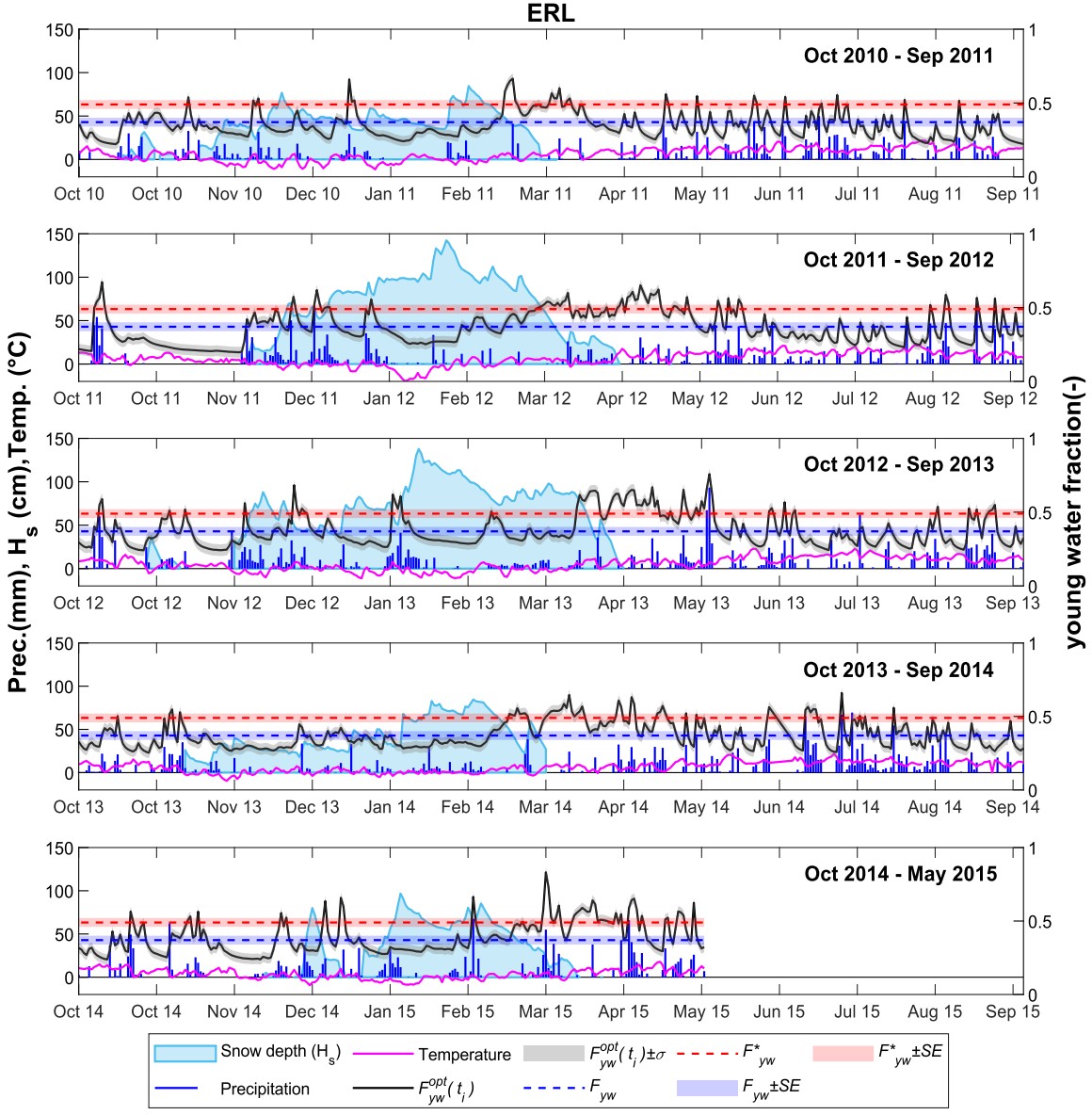

**Figure 7** Time series of daily precipitation, snow depth, air temperature and $F_{yw}^{opt}$ for the ERL catchment. Each panel reports a different hydrologic year.

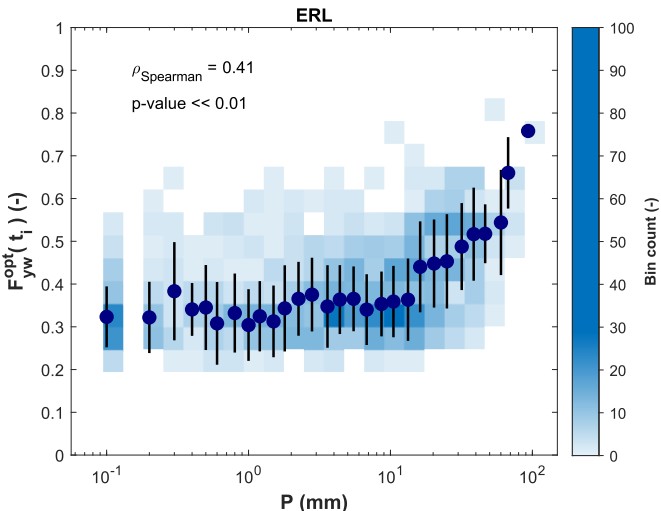

**Figure 8** Correlation between daily $F_{yw}^{opt}$ and daily precipitation when precipitation is higher than 0. Blue points indicate the median $F_{yw}^{opt}$ observed in the stream corresponding to different ranges of daily precipitation with error bars indicating the standard deviation. These median $F_{yw}^{opt}$ are plotted against the median daily precipitation in each range. The blue intensity of the bins indicates the number of observations within each bin. A rapid increase in young water fraction is observed when the daily precipitation is about 10 mm/d, thus reflecting hydrological connectivity and the generation of rapid flow paths.

### 3.3 Limitations of the *EXPECT* method and recommendations for future applications

While the *EXPECT* method can offer valuable insights into the young water fraction's discharge sensitivity and its time-variability, it is not without its limitations. The assumption of considering EC as a proxy of streamwater age may not hold true in all hydrological systems. For example, human activities, such as mining, irrigation or wastewater inputs can alter the streamwater EC in unpredictable ways. Another example involves catchments with highly soluble rocks in the aquifers (e.g., limestone or gypsum), that are susceptible to dissolution by water. It has been shown that EC can increase with $Q$ in some karst systems due to remobilization of the circulating water in the fractured areas (Balestra et al., 2022). Therefore, the $F_{yw}$-EC relationship (Eq. 3) can be very different from that in our three study catchments that are mainly groundwater influenced. Indeed, also an early study advised to be mindful of EC behaviour since it depends on specific characteristics of each catchment (Laudon and Slaymaker, 1997). Accordingly, for future applications of the method presented in this paper, we recommend to start visualizing the relationship between flow-specific young water fractions and flow-specific electrical conductivities with the aim of constructing a site-specific mixing relationship, as suggested by Kirchner (2016b). Please, note that this relationship could be potentially different from an exponential mixing model. Indeed, the use of the exponential mixing model is not pretended to be the definitive answer to the problem of choosing the right mixing model for flow partitioning in young and old water. Accordingly, if the most suitable mixing model turns out to be different from an exponential mixing model, the equations presented in this study will need to be adapted to the specific case study. However, the method's application scheme for calibrating the endmembers can still be employed. Nevertheless, in some catchments

with short and sparse isotope timeseries, flow-specific young water fractions cannot be estimated reliably (von Freyberg et al., 2018b). von Freyberg et al. (2018a) were able to estimate reliable flow-specific young water fractions for nine Swiss catchments that disposed of isotope timeseries 4 to 5 years-long with a minimum number of samples from 81 to a maximum of 140, where streamwater grab samples were collected approximately fortnightly. Thus, we suggest an isotope data set with these characteristics to construct a reliable site-specific mixing model with both flow-specific EC and $F^Q_{yw}$.

Another major limitation of the *EXPECT* method is its strong dependency on reliable $F_{yw}$ and $F^*_{yw}$ estimates (i.e., assumptions ii) and iii) in section 2.2). If streamwater isotope data are short or sparse, $F_{yw}$ or $F^*_{yw}$ can be highly uncertain and the EC endmembers cannot be constrained sufficiently well. Recently, Gallart et al. (2020a) revealed that by using a weekly sampling frequency, unweighted and flow-weighted young water fractions were significantly lower than results with virtual (perfect) sampling. Thus, for the same catchment, we could potentially obtain different EC endmembers if stable water isotopes were sampled at higher or lower temporal resolution. Accordingly, we strongly recommend evaluating how the uncertainty in $F_{yw}$ or $F^*_{yw}$ propagates in the uncertainty of the calibrated endmembers as described in section 2.2.

For many catchments, $Q$ and EC values are measured at sub-hourly resolution. Thus, theoretically the *EXPECT* method could provide reasonable young water fraction estimates results at these resolutions as well. However, we should consider that short-term variations in EC may not necessarily represent short-term variations in water age. For example, Calles (1982) showed for a small stream in Sweden that diurnal variations in EC seem to be due to evapotranspiration, but also the influence from gravity variations may play a role. Moreover, a past study in a pre-alpine river in Switzerland revealed that diurnal fluctuation of EC can be due to biogeochemical processes, such as calcite precipitation and photosynthesis (Hayashi et al., 2012). Accordingly, the biological (photosynthesis and respiration) and chemical processes (carbonate equilibrium and calcite precipitation) can play a key role in controlling $Ca^{2+}$ and $HCO^-_3$ concentrations and, consequently, EC (Nimick et al., 2011; Hayashi et al., 2012). By calculating the average daily EC, thus removing diurnal and nocturnal EC dynamics, it should better reflect variations in water age under the *EXPECT* method assumptions. Accordingly, we recommend applying the method by using daily mean of EC.

## 4    Summary and Conclusions

The discharge sensitivity of the young water fraction ($S^*_d$) is a useful metric that quantifies how the proportion of streamflow younger than 2–3 months changes as a catchment becomes wetter. In a past study, $S^*_d$ was obtained by fitting a sine-function to the streamwater isotope values, assuming an exponential relationship between young water fraction and discharge (Gallart et al., 2020b). Most available streamwater isotope datasets are characterized by a relatively low sampling frequency, which often fail to capture the entire flow regime from very low to very high discharges. This can result in highly uncertain or unrealistic estimates of the discharge sensitivity of young water fractions. Therefore, this paper aims at incorporating EC and $\delta^{18}O$ data to develop a new method that a) estimates young water fractions at high temporal resolution by taking advantage of continuous EC measurements, and that b) better constrains the estimated discharge sensitivity.

We have designed the *EXPECT* method which combines the sine-wave model of the seasonal isotope cycles and an alternative EC-based hydrograph separation. Specifically, we use an exponential mixing model in which EC endmembers are calibrated by using unweighted and flow-weighted young water fractions obtained from $\delta^{18}O$ data. By considering the calibrated endmembers, daily and biweekly (sampling) young water fractions are estimated by using EC measurements considered as a proxy of the water age. The *EXPECT* method was tested in three small experimental catchments in Switzerland.

The application of this multi-tracer method has revealed that the optimal EC endmembers lie beyond the range of measured EC in streamwater. This result reflects that streams are commonly a mixture of young and old water and that corresponding EC endmembers are difficult to be obtained experimentally. The discharge sensitivities of the young water fractions obtained with the *EXPECT* method agree well with those obtained with the conventional approach that uses only isotope data. However, the *EXPECT* method significantly reduced the standard error of discharge sensitivity. In addition, the method allows for estimating young water fractions at daily resolution, which provides interesting insights into short-term variations of streamwater age with changes in meteorological conditions, e.g., during snow accumulation and snowmelt. Young water fractions at biweekly (i.e., sampling) resolution also revealed high reliability, thus highlighting the general applicability of this method also in ungauged catchments: $\delta^{18}O$ and EC data can be both obtained from laboratory analysis of collected water samples while $Q$ can be directly measured in the stream during sampling dates with conventional methods (e.g., current meter method, weir method) without the presence of fixed instrumentation for measuring $Q$ and EC.

To conclude, a recent review paper (Benettin et al., 2022) highlighted the challenge of integrating non-conservative tracers in lumped models due to a missed definition of catchment-scale chemical properties. Overall, the EC resulted for the three study catchments as an informative property that keeps track in an integrated way of faster (younger) and slower (older) flow paths at the catchment scale. Considering the necessary precautions regarding the use of EC, the methodology presented in this paper can be applied (with possible adaptations) to other catchments to generate new insights into transit times, hydrologic flow paths and related sources.

**Appendix A: Limitations of the linear mixing model**

In order to use EC to separate the hydrograph into young and old water at a specified time $t_i$, we may employ the 2-component EC-based Hydrograph Separation (ECHS), built on the water (Eq. A1) and tracer (Eq. A2) mass balance:

$$F_{yw}(t_i) + F_{ow}(t_i) = 1 , \tag{A1}$$

$$EC(t_i) = EC_{yw}F_{yw}(t_i) + EC_{ow}F_{ow}(t_i) , \tag{A2}$$

Where, $EC(t_i)$ is the electrical conductivity measured in the stream at the time $t_i$, $EC_{yw}$ is the young water EC endmember, $EC_{ow}$ is the old water EC endmember. By solving the system of two equations (Eq. A1 and Eq. A2) with two variables ($F_{yw}(t_i)$ and $F_{ow}(t_i)$ ), we can obtain the explicit expression of $F_{yw}(t_i)$ :

$$F_{yw}(t_i) = \frac{EC(t_i) - EC_{ow}}{EC_{yw} - EC_{ow}},$$ (A3)

As mentioned in section 2.2, we assume $EC_{yw} < EC_{ow}$. However, by performing the constrained ECHS (section 2.2) in which the two endmembers ($EC_{yw}$ and $EC_{ow}$) are calibrated, the optimization algorithm finds $EC_{yw} = 0$, that is exactly the lower bound of the defined range $[0, +\infty)$ in which the optimization algorithm searches the solution. This result suggests that the algorithm wants to search the best solution below the lower bound of the specified range, thus potentially returning a negative $EC_{yw}$ value. This is consistent with the negative $EC_{yw}$ obtained by fitting a linear model on median EC vs $F^Q_{yw}$ of the three study catchments (Fig. 3). Obviously, this mathematical solution is not physically acceptable, but we can investigate this result to better understand the catchment functioning. Accordingly, if we make explicit $EC(t_i)$ from Eq. (A3), we find a linear decrement of $EC(t_i)$ with the increasing $F_{yw}(t_i)$ (Eq. A4):

$$EC(t_i) = (EC_{yw} - EC_{ow})F_{yw}(t_i) + EC_{ow} = \alpha\, F_{yw}(t_i) + EC_{ow},$$ (A4)

By requiring a negative $EC_{yw}$ as best solution, the constrained ECHS suggests that, for an exhaustive description of the catchments behaviour, $EC(t_i)$ needs to rapidly decrease at low $F_{yw}(t_i)$, as shown by the red lines in Fig. 9a. Nevertheless, physical reasons limit the slope ($\alpha$) of this line ($\alpha \geq -EC_{ow}$); the most extreme, but still acceptable condition (i.e., when $EC_{yw} = 0$ and $\alpha = -EC_{ow}$) is indicated by the dashed black line in Fig. 9a. Accordingly, to obtain a rapid decrease of $EC(t_i)$ at low $F_{yw}(t_i)$, but maintaining positive $EC_{yw}$, it is necessary to improve the linear mixing model. As visible from Fig. 9b, the exponential mixing model described in section 2.2 resulted suitable to describe a rapid decrease of $EC(t_i)$ at low $F_{yw}(t_i)$ by maintaining a positive $EC_{yw}$.

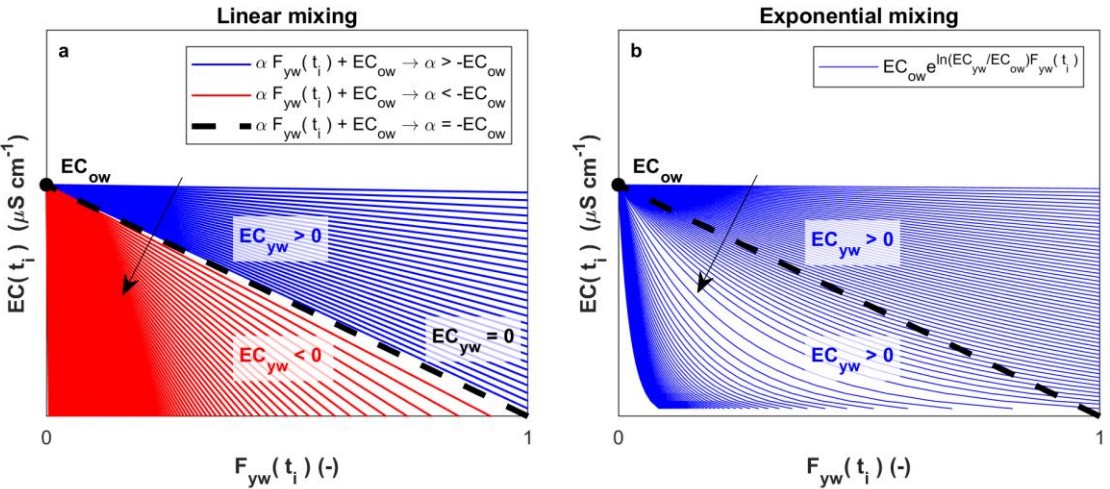

**Figure 9 a) Limits of the linear decay of $EC(t_i)$ with increasing $F_{yw}(t_i)$. Red lines with slope ($\alpha$) lower than $-EC_{ow}$ are not physically admitted since they imply a negative $EC_{yw}$; b) the exponential mixing overcomes this limit. Black arrows indicate the direction in which $EC_{yw}$ decreases.**

**List of symbols**

| | |
|---|---|
| $*$ | *Indicates a flow-weighted variable* |
| $a$ | *Parameter of the exponential mixing model reported in Eq. (3)* |
| $A_P$ | *Precipitation isotope cycle amplitude (‰) obtained through a volume-weighted robust fit of a sine function on the isotopic composition of precipitation.* |
| $A_S$ | *Streamwater isotope cycle amplitude (‰) obtained through robust fit of a sine function on the isotopic composition of streamwater* |
| $A^*_S$ | *Streamwater isotope cycle amplitude (‰) obtained through a flow-weighted robust fit of a sine function on the isotopic composition of streamwater.* |
| $A^*_S(Q)$ | *Streamwater isotope cycle amplitude (‰) varying with discharge: $A^*_S(Q) = A_P F^*_{yw}(Q)$* |
| $A^Q_S$ | *Streamwater seasonal isotope cycles amplitude (‰) obtained by fitting sinusoids separately to the isotope data collected in different discrete flow regimes as described in Kirchner (2016b) and von Freyberg et al. (2018a).* |
| $C_P$ | *Isotopic composition of precipitation (‰)* |
| $C_S$ | *Isotopic composition of streamwater (‰)* |
| $DR$ | *Daily resolution* |
| $EC$ | *Electrical conductivity ($\mu S\ cm^{-1}$)* |
| $EC(t_i)$ | *Electrical conductivity ($\mu S\ cm^{-1}$) in streamwater at the generic time $t_i$* |
| $EC_{yw}$ | *Young water electrical conductivity endmember ($\mu S\ cm^{-1}$)* |
| $EC_{yw}^{raw}$ | *First-order estimate of young water electrical conductivity endmember ($\mu S\ cm^{-1}$) obtained evaluating the exponential model, fitted on the median flow-specific EC vs $F^Q_{yw,}$ for $F^Q_{yw} = 1$* |
| $EC_{yw}^{opt}$ | *Optimized young water electrical conductivity endmember ($\mu S\ cm^{-1}$) derived from calibration* |
| $EC_{ow}$ | *Old water electrical conductivity endmember ($\mu S\ cm^{-1}$)* |
| $EC_{ow}^{raw}$ | *First-order estimate of old water electrical conductivity endmember ($\mu S\ cm^{-1}$) obtained evaluating the exponential model, fitted on the median flow-specific EC vs $F^Q_{yw}$, for $F^Q_{yw} = 0$* |
| $EC_{ow}^{opt}$ | *Optimized old water electrical conductivity endmember ($\mu S\ cm^{-1}$) derived from calibration* |
| $ECHS$ | *EC-based Hydrograph Separation* |
| $ERL$ | *Erlenbach catchment* |
| $EXPECT$ | *Two-component Electrical Conductivity-based hydrograph separaTion employing an EXPonential mixing model.* |
| $f$ | *Frequency of the seasonal cycle (equal to 1 $y^{-1}$ for a seasonal cycle)* |
| $F_{yw}$ | *Unweighted average isotope-based young water fraction (-) obtained as $A_S/A_P$* |

| | | |
|---|---|---|
| | $\widetilde{F}_{yw}$ | *Unweighted average HS-based young water fraction (-) obtained with the exponential mixing model.* |
| 605 | $F^*_{yw}$ | *Flow-weighted average isotope-based young water fraction (-) obtained as $A^*_S/A_P$* |
| | $\widetilde{F}^*_{yw}$ | *Flow-weighted average HS-based young water fraction (-) obtained with the exponential mixing model.* |
| | $F^*_{yw}(Q)$ | *Young water fraction (-) varying with discharge following the exponential-type equation of Gallart et al. (2020b)* |
| 610 | $F_{yw}(t_i)$ | *Young water fraction (-) at the generic time $t_i$* |
| | $F_{ow}(t_i)$ | *Old water fraction (-) at the generic time $t_i$* |
| | $F^{opt}_{yw}$ | *Optimized young water fractions (-), obtained with the exponential mixing model, by using the calibrated endmembers $EC^{opt}_{yw}$ and $EC^{opt}_{ow}$* |
| | $F^{opt}_{yw}(t_i)$ | *Optimized young water fraction (-) at the generic time $t_i$, obtained with the exponential mixing model, by using the calibrated endmembers $EC^{opt}_{yw}$ and $EC^{opt}_{ow}$* |
| 615 | | |
| | $F^*_0$ | *Virtual young water fraction (-) when $Q = 0$ in the exponential-type equation of Gallart et al. (2020b).* |
| | $F^{EXP}_0$ | *Virtual young water fraction (-) when $Q = 0$ obtained by fitting Eq. (6) of Gallart et al. (2020b) on $F^{opt}_{yw}$ vs $Q$ data.* |
| 620 | $F^Q_{yw}$ | *Young water fractions (-) estimated in discrete flow regimes as described in Kirchner (2016b) and von Freyberg et al. (2018a): $F^Q_{yw} = A^Q_S/A_P$* |
| | HS | *Hydrograph Separation* |
| | $H_S$ | *snow depth (cm)* |
| | ID | *Identifier* |
| 625 | IHS | *Isotope Hydrograph Separation* |
| | IQR | *Interquartile range* |
| | k | *Number (-) of pecipitation isotope samples* |
| | $k^*_S$ | *Constant (‰) representing the vertical offset of the seasonal cycle.* |
| | LUE | *Lümpenenbach catchment* |
| 630 | n | *Number (-) of Q and EC observations* |
| | m | *Number (-) of streamwater isotope samples* |
| | MTT | *Mean Transit Time* |
| | obj | *Objective function* |
| | Old water | *Water with transit times roughly higher than 2-3 months (definition given in this paper)* |

| | | |
|---|---|---|
| 635 | $P$ | Precipitation (mm) |
| | $P(t_k)$ | Volume of precipitation (mm) used for the volume-weighted fit of precipitation isotopes. |
| | $q_1$ | $1^{st}$ quartile |
| | $q_2$ | $2^{nd}$ quartile |
| | $q_3$ | $3^{rd}$ quartile |
| 640 | $Q$ | Discharge (mm d$^{-1}$) |
| | $Q_{med}$ | Median stream discharge (mm d$^{-1}$) |
| | $Q(t_i)$ | Discharge at the time $t_i$ (mm d$^{-1}$) |
| | $Q(t_j)$ | Discharge at the time $t_j$ (mm d$^{-1}$) |
| | $Q_{50/50}$ | Median discharge (mm d$^{-1}$) at which $50 \pm 1\%$ of both young and old water exist in streamflow. |
| 645 | Young water | Water with transit times roughly lower than 2-3 months (definition given in this paper) |
| | $S^*_d$ | Discharge sensitivity of young water fraction (d mm$^{-1}$) obtained with the method of Gallart et al. (2020b) |
| | $S_d^{EXP}$ | Discharge sensitivity of young water fraction (d mm$^{-1}$) obtained by fitting Eq. (6) of Gallart et al. (2020b) on $F_{yw}^{opt}$ vs $Q$ data. |
| 650 | SE | Standard error |
| | SR | Sampling resolution |
| | $t_i$ | Generic time in which $Q$ and EC are measured. |
| | $t_j$ | Generic time in which a streamwater isotope sample is collected. |
| | $t_k$ | Generic time in which a precipitation isotope sample is collected. |
| 655 | VOG | Vogelbach catchment |
| | $\delta^2 H$ | Isotope content (‰) considering deuterium. |
| | $\delta^{18} O$ | Isotope content (‰) considering oxygen-18. |
| | $\delta^{18} O_p$ | Isotope content of precipitation (‰) considering oxygen-18. |
| | $\delta^{18} O_P(t_k)$ | Isotope content of precipitation (‰) at the time $t_k$ considering oxygen-18. |
| 660 | $\delta^{18} O_S$ | Isotope content of streamwater (‰) considering oxygen-18. |
| | $\delta^{18} O_S(t_i)$ | Isotope content of streamwater (‰) at the time $t_i$ considering oxygen-18. |
| | $\varphi^*_S$ | Phase of the seasonal cycle (rad) |
| | $\sigma$ | Standard deviation |
| | $\rho_{Spearman}$ | Spearman correlation coefficient |

665

*Data availability*. Time series of $\delta^{18}O$ in streamflow and precipitation for the ERL, LUE and VOG catchments are available in the data repository Zenodo at https://zenodo.org/record/4057967#.Y00oMHZBxPY (Staudinger et al., 2020). Daily discharge and electrical conductivity data for the ERL, LUE and VOG catchments are available from the *Swiss Federal Institute for Forest, Snow and Landscape research* (*WSL*) data portal at https://www.envidat.ch/#/metadata/longterm-hydrological-observatory-alptal-central-switzerland . The shape files (.shp) of the ERL, LUE and VOG catchments are available at https://zenodo.org/record/4057967#.Y00oMHZBxPY (Staudinger et al., 2020).

*Author contributions*. AG, JvF, and SF identified the research gap, designed the *EXPECT* method and prepared the paper. AG implemented the *EXPECT* method in a Matlab code with the support of DG and DC. All authors revised the manuscript and gave final approval to the submitted version.

*Competing interests*. The authors declare that they have no conflict of interest.

*Acknowledgements*. This work was supported by the "*PRIN MIUR 2017SL7ABC_005 WATZON Project*" and by *"Funding 2021 Fondazione CRT"*. This publication is part of the project NODES which has received funding from the MUR –M4C2 1.5 of PNRR with grant agreement no. ECS00000036. Jana von Freyberg was supported by the Swiss National Science Foundation SNSF (grant PR00P2_185931). The results of this study have been discussed within the COST Action "WATSON", CA19120, supported by COST (European Cooperation in Science and Technology).

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
