# Peer review of "Technical Note: two-component *Electrical Conductivity-based* hydrograph separa*T*ion employing an *EXP*onential mixing model (*EXPECT*) provides reliable high temporal resolution young water fraction estimates in three small Swiss catchments"

_EGUsphere, 2023_

## Referee Comment (RC3)

Dear authors,

I would like to thank you for the effort put into addressing my comments. The discussion is the best way to clarify the ideas and realize possible misunderstandings or drawbacks. Our discussion can be perhaps useful also to journal readers.

1. I understood that you did not do hydrograph separation with stable isotopes. It is not necessary to to rewrite lines 165-172. The reader can obtain a more detailed information from your response to my comments.

2. The key assumption of your approach is the exponential relationship between EC and young water fraction. Could you try to justify it also in some other way than just mathematically (l. 176-190)?

3. I have downloaded and checked the discharge and EC data for your catchments. Some thoughts are given below (you do not need to respond to them). Although I am still not convinced about the use of EC, the manuscript describes the proposed approach clearly.

4. You may think about using the list of symbols, because there are many symbols from earlier works and some other symbols used in your study. Such a list might be helpful to someone who is not so familiar with all the literature and would like to use your method.

5. It is clear that "old water" in your study is related to the young water fraction (the metric calculated from seasonal isotope variability); i. e. "old water"=1-young water fraction. However, this term is the same as the "old water" from the isotopic hydrograph separation conducted by a mixing formula. To avoid the confusion, it may be useful to explain, e.g. in the List of symbols that your "old water" is different.

6. Despite my comments on the manuscript, if the editor and other reviewer(s) decide that the manuscript can be published, I will not have a problem to accept such a decision.

7. I agree that you acknowledged many uncertainties related to the use of the method. What I mind is this:

   A. We (the hydrological community) know for decades that determination of the input (tracer concentration of the water entering the system, e.g. a catchment) is uncertain. The composition of water infiltrating into the soil that eventually appears in the output (e.g. in catchment runoff) is almost always unknown. We acknowledge this uncertainty and use tracer content in precipitation, because that is what we can (more easily) measure and in sometimes adjust it using different approaches.

   B. We know that tracer variability in the input varies both temporally and spatially. The range of temporal variability differs in different years. We acknowledge this uncertainty and approximate the input concentration by the sine curve having the same amplitude over different years. Spatial variability in larger catchments is often neglected.

   C. Several approaches are used to estimate the sine curve's amplitude (limiting or accepting the outliers) for weighted or unweighted data. Study periods are sometimes shorter than several years. All this brings the uncertainty which we acknowledge and determine the amplitude.

   D. From the amplitudes we calculate the metric (an exact number) characterizing studied system. For many years it was the mean residence/transit time. After the inspiring work by Kirchner (2016) we prefer to use the metric called young water fraction.

   E. Young water fraction (an exact number) is defined as "the fraction of runoff with transit times of less than roughly 0.2 years" (Kirchner, 2016). It represents an average over the study period. It

seems obvious that when the discharge in a study catchment increases, the young water fraction should likely be greater than in the low flow periods when the streamflow is supplied by water that probably stayed in the catchment longer (we do not know how much longer than 2-3 months, but part of that water may be in the catchment not much longer 2-3 moths, i. e. 4, 5, 6?).

F.  We introduce a new metric called discharge sensitivity of the young water fraction and assume the exponential relationship between the young water fraction and a virtual young water fraction for discharge equal to zero.

G.  It is fascinating and potentially very useful to know how big is the young water fraction on every day, hour, etc.  We continue with the development of methodology and calculate daily young water fractions using another, non-conservative tracer (EC) and two-component hydrograph separation. We estimate the unknown tracer concentrations for the two end members though calibration. We assume that there is exponential relationship between the tracer and young water fraction and optimize the daily values so that their average is the same as the young water fraction obtained from seasonal variations of stable isotopes. We acknowledge possible uncertainties.

H.  Having the daily young water fractions, we can investigate their relationships with meteorological drivers, and so on and so forth. …..

I.  A to H indicate that we are adding uncertainties with every step in the development of our methodology. Please note I am saying "adding" not "accumulating", because I do not know if the uncertainty increases in the described chain of methodology development.

J.  We are acknowledging the uncertainty, but continuing to develop the methodology and adding other uncertainties. The result is that since the 1970'/1980' we moved from a simple method providing a rough, but useful characteristic (especially in groundwater hydrology, because it matters if possible pollutant enters an aquifer with mean transit time 6 or 26 months for example) to a complex methodology involving many acknowledged uncertainties providing "exact" numbers for the short time steps.

K.  I am not sure how much can the obtained numbers be trusted and whether we are obtaining a substantially new knowledge about the subject of our study, e. g. catchment hydrological cycle (in addition to the information on tracer dynamics). Benetin et al (2022) noted:  "In the light of the complexity of the theoretical apparatus underlying time-variant TTDs …, one might wonder if this effort is actually worthwhile and all this complexity is really needed for practical purposes. Our claim is that, while time-variance might not be needed a priori to characterize transport processes in a catchment, it directly affects tracers and solute signals in stream water and plant water. Therefore, acknowledging and incorporating this time variance may be necessary to capture and explain both high-frequency and long-term tracer dynamics."

I have downloaded the discharge and EC data from your catchments and period October 1st, 2010-November 30th, 2015 which is approximately your study period according to Table 1.

1.  I agree with you that discharge increase almost always corresponds to EC decrease and vice versa.

2.  A few thoughts on the optimized EC values of the endmembers: The low flow periods in the study catchments are never very long (even in winter). Yet, the difference between the

optimized EC of the old water fraction in ERL (501 $\mu$S.cm$^{-1}$) and the minimum EC values measured in the stream in period October 2010-November 2015 (334.3 $\mu$S.cm$^{-1}$) is quite high. Even the absolute EC minimum in ERL (439.5 $\mu$S.cm$^{-1}$) between January 1978 and February 2023 (daily data) that was measured on 23$^{rd}$ January 1990, i.e. outside of your study period, was quite different from the optimized value. I am therefore not sure if the optimized EC values are correct. The young water fraction was maybe not very big in January 1990 at catchment discharge of about 0.3 l.s$^{-1}$. I would assume that streamflow EC would be closer to that of the groundwater, i.e. the measurements over long periods could identify this end member. Similarly, the optimized EC values of the young water fractions seem to be a little higher than data on Central European precipitation suggest (Monteith et al., 2023), but it can be argued that the young water fraction contains some soil water with higher EC.

3. According to coefficient of determination, Q explains about 50% of daily EC variability in your catchments. It would be great if part of the variability could be explained by young water fraction. However, how can it be confirmed or rejected if daily young water fractions were estimated on the basis of EC?

References:

Benettin, P., Rodriguez, N. B., Sprenger,M., Kim, M., Klaus, J., Harman, C. J.,et al. 2022. Transit time estimation in catchments: Recent developments and future directions. Water Resources Research, 58, e2022WR033096. https://doi.org/10.1029/2022WR033096

Monteiths, D. T., Henrys, P. A., Hruška, J., de Witt, H. A., Krám, P., Moldan, F., Posch, M., Räike, A., Stoddard, J. L., Shilland, E. M., Pereira, G. M., Evans, Ch., D. 2023. Long-term rise in riverine dissolved organic carbon concentration is predicted by electrolyte solubility theory. Sci. Adv. 9 (3), eade3491. DOI: 10.1126/sciadv.ade3491

---

## Author Comment (AC1)

**Dear Anonymous Referee #1,**

**We thank you for your appreciation about our effort in developing a multi-tracer method**
**for advancing the knowledge of young water in catchment hydrology. We thank you for**
**your comments, also if, in our opinion, the method has not been fully understood.**

**We think that many of the criticisms made should be supported by scientific results in**
**the literature to bring substance to the discussion. In this regard we answered point-by-**
**point to your comments by citing published works that support our assumptions and**
**methodology.  Moreover, many problems that have been listed have already been**
**addressed by the authors along the manuscript.**

**Perhaps, we will have to clarify some points better and we will be glad to do so in the**
**revised version of the manuscript if we receive a positive editor's response.**

**Accordingly, we kindly ask to reconsider the decision of not-recommending the**
**manuscript for publication in light of our answers.**

**With kind regards,**

**The Authors**

1. Hydrograph components, i.e. the event and pre-event water, do not represent young
and old water fractions defined by the young water fraction concept.

**Please, note that we are not subdividing the hydrograph into event water and pre-event**
**water. This can be done by applying a time-source separation with the only use of stable**
**water isotopes as tracer (Klaus and McDonnell, 2013). This cannot be done with**
**geochemical tracers like ECs.**

**What we actually do is partially described in Kirchner (2016b):**

*"The young water fraction $F_{yw}$ may also be helpful in inferring chemical processes*
*from streamflow concentrations of reactive chemical species. Many reactive species*
*exhibit clear concentration–discharge relationships. Because one can determine how*
*$F_{yw}$ varies, on average, across different ranges of discharge, one can potentially*
*construct mixing relationships between $F_{yw}$ and the concentrations of reactive*
*species. If the measurable range of $F_{yw}$ is wide enough, one may even be able to*
*estimate the end-member concentrations corresponding to idealized "young water"*
*($F_{yw}$ = 1) and "old water" ($F_{yw}$ = 0)"*

**Accordingly, what we did is to construct a mixing relationship between $F_{yw}$ and the**
**concentrations of reactive species (integrated in the ECs measure): the exponential decay**
**of ECs with increasing $F_{yw}$ (see Eq. 5 of the preprint). From this mixing relationship, we**
**can separate the hydrograph in young water and old water by choosing appropriate end-**

**members. If possible, one could potentially measure the EC of young water (< 2-3 months)**
**and old water (> 2-3 months) in the stream when flow specific young water fractions are**
**equal to 1 or 0, respectively (Kirchner, 2016b). Since the flow conditions in which $F_{yw} = 1$**
**or $F_{yw} = 0$ are rare scenarios (see line 189-193 of the preprint), we obtain the two EC end-**
**members through calibration because we have additional information deriving from**
**stable water isotopes that we can use to constrain the two end-members: the flow-**
**weighted average young water fraction ($F^*_{yw}$) and the time-weighted average young water**
**fraction ($F_{yw}$) (both obtained with the amplitude ratio approach). The calibrated optimal**
**$EC_{ow}$ and $EC_{yw}$ we have obtained are respectively higher and lower than the maximum**
**and minimum $EC_S$ measured in the stream, thus suggesting that they are the idealized old**
**water ($F_{yw} = 0$) and young water ($F_{yw} = 1$) EC endmembers (see lines 256-263 of the**
**preprint).**

**Another way to obtain the endmembers, by strictly following Kirchner (2016b), is to look**
**at how flow-specific young water fractions vary with the selected tracer ($EC_S$ in this case)**
**measured in the same flow-regime. We report hereafter flow-specific $EC_S$ (median values)**
**against flow-specific young water fractions for the three study catchments (Fig. 1). First,**
**we observe a non-linear relationship that confirms the mixing relationship presented in**
**the Eq. (5) of our manuscript (i.e., an exponential decrease of $EC_S$ with increasing $F_{yw}$).**
**Second, by fitting an exponential model on data we can obtain the end-members ($EC_{ow}$**
**and $EC_{yw}$) values evaluating the $EC_S(F_{yw})$ equation for $F_{yw} = 0$ and $F_{yw} = 1$. The main**
**problem is that these end-members would be highly uncertain since the curve is poorly**
**constrained at very low and very high $F_{yw}$ (see Fig. 1).**

**We did not insert this analysis in the manuscript, but we are glad to add it in the revised**
**version to better clarify our method.**

[Figure]

[Figure]

$$EC_S = 362.13 \exp(-2.76 F_{yw})$$

**Fig.1** *Median flow-specific $EC_S$ against flow-specific $F_{yw}$ for the three study catchments. Horizontal bars indicate*
*the flow-specific $F_{yw}$ standard error. The black curve indicates the exponential fit, while the dashed black lines*
*indicate the 90% prediction bounds.*

**In summary, we do not separate the hydrograph by using the stable water isotopes (thus**
**obtaining event water and pre-event water). We separate the hydrograph by using $EC_S$**
**and by calibrating the EC end-members to reflect the age of the two components (young**
**water and old water). We can rewrite lines from 165 to 172 of the preprint to clarify this**
**point better.**

**In light of this clarification, we think that your first comment is derived from a**
**misunderstanding of the hydrograph separation we actually did and how we have**
**operated to perform it. Of course, event and pre-event water do not represent young and**
**old water. Event water is a portion of young ($<$ 2-3 months) water while old ($>$ 2-3 months)**
**water is a portion of pre-event water.**

2.  The conceptual problem in my opinion is that if the young water fraction is defined as
a characteristic (metric) calculated from the seasonal variability of a tracer, it is attributed
to the seasonal time scale. Thus, it is not meaningful to "improve" the temporal resolution
of such a characteristic for finer scales (hydrographs) if the tracer streamflow data is
sparse.

**There is no conceptual problem in our method. We have obtained a young water fraction**
**timeseries (at daily or sampling resolution) so that its (unweighted) average is equal to**
**$F_{yw}$ and its flow-weighted average is equal to $F^*_{yw}$ and these are the definitions of $F_{yw}$ and**
**$F^*_{yw}$ obtained from seasonal tracer cycles. See the quote from Kirchner (2016b):**

*"Flow-weighted fits to the seasonal tracer cycles accurately predict the flow-weighted*
*average $F_{yw}$ in streamflow, while unweighted fits to the seasonal tracer cycles*
*accurately predict the unweighted average $F_{yw}$"*

**Of course, the young water fraction is calculated from seasonal tracer cycles as a**
**characteristic at the seasonal time scale. However, "the fraction of runoff with transit**

**times of less than roughly 2-3 months (Kirchner, 2016a)" is a definition that could be applied also to finer scales. Since the young water fraction calculated from seasonal tracer cycles is an average value (Kirchner 2016a, b) over the period of isotope sampling, it depends on the young water fractions we have in the hydrograph at each (finer) time-step over the period of isotope sampling. Accordingly, it is very useful to improve the temporal resolution of young water fraction estimation: this gives additional information on the hydrological processes and conditions occurring in a catchment that cannot be perceived at the seasonal time scale and that can potentially explain the low or high average young water fraction value.**

**Moreover, we would not say that "is meaningful to improve the temporal resolution of such a characteristic for finer scales" since previous papers in the scientific literature have addressed the problem of understanding how the young water fraction varies with different flow regimes, and thus over time (e.g. by using automatic water samples triggered by time and by flow, see Gallart et al. 2020a, 2020b). In this regard, von Freyberg et al. (2018) and, subsequently, Gallart et al. (2020b) worked on the sensitivity of young water fraction to discharge ($Q$), the latter considered as a proxy of the catchment wetness. By studying how the young water fraction changes with $Q$ is implicitly an investigation of how the young water fraction varies over time since $Q$ varies over time. In this regard, we report here a quote of von Freyberg et al. (2018) paper: "*In individual catchments, one would also expect young water fractions (and thus seasonal isotope cycles) to be variable in time, i.e. to be larger during periods of stronger precipitation forcing and wetter antecedent conditions, as shallower, faster flow paths become more dominant, and as the stream network extends farther into the landscape, shortening the average path length of subsurface flow (Godsey and Kirchner, 2014).*"**

**Please, note that the method used to estimate the so-called discharge sensitivity of young water fraction has been designed to be applied in catchments in which tracer streamflow data is sparse. In this regard, again, we report here a quote of von Freyberg et al. (2018) paper: "*…we calculated the linear slope of the relationship between Q and $F_{yw}$, using a method that does not require breaking the streamwater isotope time series into separate flow regimes (and thus has more modest data requirements…)*"**

3. **\*EC is not a conservative tracer** although it is generally true that the longer the water stays in the catchment, the more ions it contains. However, EC varies also with air (water) temperature despite temperature compensation employed in the sensors and if an event water meets very soluble minerals, it can have high EC as well.

**About the combined use of not conservative tracers and the young water fraction, please see, again, the quote from Kirchner (2016b):**

> "*The young water fraction $F_{yw}$ may also be helpful in inferring chemical processes from streamflow concentrations of reactive chemical species. Many reactive species exhibit clear concentration–discharge relationships. Because one can determine how*

*$F_{yw}$ varies, on average, across different ranges of discharge, one can potentially*
*construct mixing relationships between $F_{yw}$ and the concentrations of reactive*
*species. If the measurable range of $F_{yw}$ is wide enough, one may even be able to*
*estimate the end-member concentrations corresponding to idealized "young water"*
*($F_{yw}$ = 1) and "old water" ($F_{yw}$ = 0)".*

**\*According to Kirchner (2016b), we do not need a conservative tracer to construct a**
**mixing relationship with $F_{yw}$. Thus, we do not understand why you think it is incorrect to**
**use ECs.**

**Of course, the use of ECs has some limitations, but we have been extremely transparent**
**about this: we have dedicated the whole section 3.3 of the manuscript regarding the**
**limitations of the *EXPECT* method (including the limitations of ECs as a tracer).**
**Accordingly, the application of the *EXPECT* method, as it is, could be critical in some**
**catchments with very soluble minerals (please, note that we have underlined this**
**limitation in the paper, see lines 353-357 of the preprint), but could be successful in many**
**other catchments (like the three alpine catchments investigated in this work).**

**In conclusion, we think that is crucial to underline the limitations of ECs as a tracer (as**
**we did in the section 3.3 of the preprint) to separate the young water from the old water,**
**but that it is wrong to say that it cannot be used since it is not a conservative tracer.**

**We can integrate in section 3.3 of the manuscript the influence of air (water) temperature**
**on ECs despite temperature compensation employed in the sensors, as you have**
**suggested. Thank you for this.**

4. **Exponential relationship\*** between EC and discharge (increasing **event water\*\***
contribution) appears when the long-term data (combining many events) is analysed.
However, **measurements I have seen\*\*\*** did not document such a relationship for
individual runoff events (hydrographs). While generally the EC does decrease with
increasing discharge for many events, it does not happen so for each event. In fact, **in**
**my experience a decrease in EC with increase in discharge was evident only during**
**larger events\*\*\*\***. Thus, it is in my opinion incorrect to apply the general relationship
emerging from the long-term data for all individual events.

**\*If possible, please indicate the equation number of the relationship you are referring to**
**since it is unclear. If you are referring to Eq. (5), please note that ECs($t_i$) decreases**
**exponentially with increasing $F_{yw}(t_i)$ (since we construct a mixing relationship between**
**$F_{yw}(t_i)$ and ECs($t_i$), Kirchner, 2016b), but we do not have discharge in the equation.**
**Accordingly, we are not assuming that ECs($t_i$) decreases with increasing $Q(t_i)$. The fact**
**that $F_{yw}(t_i)$ increases with $Q(t_i)$ is a result of the paper (not an assumption) as visible in**
**Fig. 4 of the preprint (and consistent with results of Gallart et al. 2020b).**

**\*\* We are not referring to event water; we are referring to young water.**

**\*\*\*Also, we do not understand what are the measurements you have seen: are you referring to the measurements of the three catchments reported in this manuscript? Are you referring to measurements of your study catchments? Are you referring to specific works published in literature? If so, it would be nice to know the temporal resolution of measurements you are referring to (is it the same as our data?) and to see what the boundary conditions of such measurements are (e.g., latitude, longitude, hydro-climatic regime, mean annual precipitation, geology, catchment area, mean catchment elevation, influence of human activities, presence of karst systems...) in order to have some concrete material for discussion.**

**Looking at daily *Q* and *EC* data** **of our three catchments it is clear that the** **dominant functioning is a decrease in ECs($t_i$) if *Q($t_i$)*  increases and viceversa****. This is visible from Fig. 2 of the preprint, but also from an in-depth analysis we did and that we report hereafter. Looking at what happens day-by-day in our data, we calculate the number of days in which to an increase in stream discharge corresponds a decrease in ECs (i.e., Q($t_{i+1}$) - Q($t_i$) > 0 AND ECs($t_{i+1}$) - ECs($t_i$) < 0 ) OR in which to a decrease in stream discharge corresponds an increase in ECs (i.e., Q($t_{i+1}$) - Q($t_i$) < 0 AND ECs($t_{i+1}$) - ECs($t_i$) > 0 ). Our results show that in 88%, 82% and 84% of the days at ERL, LUE and VOG catchments, respectively, we observe this functioning. When this is not true, we calculate the absolute difference in ECs in the two adjacent days (i.e., ECs($t_{i+1}$) - ECs($t_i$)). The frequency distributions of these differences reveal a third quartile of 10.9, 6.8 and 4.8 μS/cm for ERL, LUE and VOG, respectively. These are little differences if compared with the range of variation of ECs in these catchments, ranging from about 80 μS/cm to 400 μS/cm.**

**According to these results, we overall agree that ECs does not decrease with increasing discharge at each time-step, but in our study catchments this happens only in a very limited number of days over a study period of about 5 years. Nevertheless, this fact does not affect the validity of Eq. (5) since we assume that ECs($t_i$) decreases exponentially with increasing $F_{yw}(t_i)$, not with increasing Q($t_i$).**

**\*\*\*\*Hydrograph separation by using ECs has been applied in a previous study where ECs decreases with increasing *Q* during both larger and smaller events (Cano-Paoli et al. 2019), and not only during larger events.**

**In conclusion, we think that it is limiting to evaluate the goodness of our method by only considering its application in catchments studied during its own experience (of which we do not have any information). Indeed, the behavior of such catchments turned out to be different from other catchments studied in literature (e.g., Cano-Paoli et al. 2019) in which the method, as it is, could potentially work well (as in the three alpine catchments of our study).**

**The applicability of the method in catchments with different characteristics (e.g.,in karst systems with very soluble minerals or in catchments where ECs increases with *Q*) is a**

**possible future development of this work: at present, we have detailed the limitations of**
**the *EXPECT* method by explaining the conditions in which it can work fine or badly.**

5.    I agree that catchments with a greater number of runoff events (i. e. more frequent
higher discharge) likely have higher young water fraction. Therefore, I appreciate the idea
of calibrating separated **event/pre-event water components\*** for all the events of period
used to determine the young water fractions to estimate the proportion of "young water
fractions" (see comment 2) in **individual events**\*\*. However, as I argue above, the EC
does not provide the reliable information. Furthermore, I think that isotopes (atoms of
elements forming water) and EC (ions of compounds saluted in water) may not provide the
same (compatible) information about the streamwater sources. Last but not least, catchment
runoff response is nonlinear. It is therefore questionable to assume that the young water
fractions of **individual events\*\*** are proportional to distribution of the event water
fractions.

\***We are calibrating the young/old water EC endmembers**

**\*\*We are not looking at specific individual events: we are estimating daily (or sampling)**
**young water fractions. In both cases we use daily $Q$ and EC$_S$ data. To work at the event-**
**scale, a finer (e.g., hourly) resolution data should be used.**

**Thank you for appreciating the idea of calibrating young/old water EC endmembers to**
**estimate the young water fractions at each time step. However, we do not agree about the**
**fact that EC$_S$ cannot provide reliable information. Of course, the use of EC$_S$ has some**
**limitations, and it should be used with care (see section 3.3 of the preprint), but our results**
**show the opposite of what you are saying. We have found consistency between the $F_{yw}(Q)$**
**relationship found by Gallart et al. 2020b (using only stable water isotopes**
**measurements) and our $F_{yw}(Q)$ relationship found by using both daily EC$_S$ measurements**
**and average young water fractions (estimated with the amplitude ratio approach) to**
**constrain the endmembers. Our results reveal a good compatibility of information.**

**Moreover, there are previous papers that conveniently used EC$_S$ and stable water**
**isotopes obtaining coherent results (e.g., Cano-Paoli et al. 2019, Mosquera et al. 2018).**
**Also, in a recent paper Riazi et al. (2022) said that water from different sources within**
**the catchment is likely to have different ages. Hence, EC can potentially provide useful**
**information on water age (lines 169-170 of the preprint).**

**Concluding, we report again the quote from Kirchner (2016b) about the synergic use of**
**young water fraction and reactive chemical species:**

  *"**The young water fraction $F_{yw}$ may also be helpful in inferring chemical processes***
  *****from streamflow concentrations of reactive chemical species.** Many reactive species***
  *exhibit clear concentration–discharge relationships. Because one can determine how*
  *$F_{yw}$ varies, on average, across different ranges of discharge, one can potentially*
  *construct mixing relationships between $F_{yw}$ and the concentrations of reactive*

*species. If the measurable range of $F_{yw}$ is wide enough, one may even be able to*
*estimate the end-member concentrations corresponding to idealized "young water"*
*($F_{yw}$ = 1) and "old water" ($F_{yw}$ = 0)".*

**Please be clearer (e.g., indicate the lines of the manuscript) about the fact that we assume**
**that "young water fractions of individual events are proportional to distribution of the**
**event water fractions". We did not make this assumption (or we do not understand what**
**you are referring to).**

**References**

Cano-Paoli, K., Chiogna, G., and Bellin, A.: Convenient use of electrical conductivity
measurements to investigate hydrological processes in Alpine headwaters, Science of The
Total Environment, 685, 37–49, https://doi.org/10.1016/j.scitotenv.2019.05.166, 2019.

Gallart, F., Valiente, M., Llorens, P., Cayuela, C., Sprenger, M., and Latron, J.: Investigating
young water fractions in a small Mediterranean mountain catchment: Both precipitation forcing
and sampling frequency matter, Hydrological Processes, 34, 3618–3634,
https://doi.org/10.1002/hyp.13806, 2020a.

Gallart, F., von Freyberg, J., Valiente, M., Kirchner, J. W., Llorens, P., and Latron, J.:
Technical note: An improved discharge sensitivity metric for young water fractions, Hydrology
and Earth System Sciences, 24, 1101–1107, https://doi.org/10.5194/hess-24-1101-2020,
2020b.

Kirchner, J. W.: Aggregation in environmental systems-Part 2: Catchment mean transit times
and young water fractions under hydrologic nonstationarity, Hydrology and Earth System
Sciences, 20, 299–328, https://doi.org/10.5194/hess-20-299-2016, 2016.

Klaus, J. and McDonnell, J. J.: Hydrograph separation using stable isotopes: Review and
evaluation, Journal of Hydrology, 505, 47–64, https://doi.org/10.1016/j.jhydrol.2013.09.006,
2013.

Mosquera, G. M., Segura, C., and Crespo, P.: Flow Partitioning Modelling Using High-
Resolution Isotopic and Electrical Conductivity Data, Water, 10, 904,
https://doi.org/10.3390/w10070904, 2018.

Riazi, Z., Western, A. W., and Bende-Michl, U.: Modelling electrical conductivity variation
using a travel time distribution approach in the Duck River catchment, Australia, Hydrological
Processes, 36, e14721, https://doi.org/10.1002/hyp.14721, 2022.

von Freyberg, J., Allen, S. T., Seeger, S., Weiler, M., and Kirchner, J. W.: Sensitivity of young
water fractions to hydro-climatic forcing and landscape properties across 22 Swiss catchments,
Hydrology and Earth System Sciences, 22, 3841–3861, https://doi.org/10.5194/hess-22-3841-
2018, 2018.

---

## Author Comment (AC2)

Dear authors,

I would like to thank you for the effort put into addressing my comments. The discussion is the best way to clarify the ideas and realize possible misunderstandings or drawbacks. Our discussion can be perhaps useful also to journal readers.

**Dear Anonymous referee #1,**

**We thank you very much for your reply to our comments (AC1) that further stimulates**

**the discussion. We are pleased to note that the discussion has solved some possible**

**misunderstandings and brought constructive comments and feedback to our manuscript.**

**Of course, this discussion will be useful to journal readers, but it is also crucial for us to**

**take a critical look at our research.**

**Please, find below a point-by-point response to your comments.**

**We will incorporate all your constructive feedback in the revised version of our**

**manuscript if we receive a positive editor's response.**

**Sincerely,**

**The Authors**

1. I understood that you did not do hydrograph separation with stable isotopes. It is not necessary to rewrite lines 165-172. The reader can obtain more detailed information from your response to my comments.

**Ok, thank you for this.**

2. The key assumption of your approach is the exponential relationship between EC and young water fraction. Could you try to justify it also in some other way than just mathematically (l.

176-190)?

**Thank you for this comment. We have realized that the exponential relationship between**

**EC and young water fraction could not appear robustly justified as presented in the**

**preprint. In this regard, we would like to incorporate the figure representing flow-specific**

**electrical conductivity vs flow-specific young water fractions (see Fig. 1 of AC1) in Section**

**2.2 (similarly to Figure 14 of Kirchner, 2016b). From this figure it is possible to visualize**

**the relationship between electrical conductivity and young water fraction. Indeed, we**

**observe that the decrease of EC with increasing young water fraction is well described by**

**the exponential model.**

3. I have downloaded and checked the discharge and EC data for your catchments. Some thoughts are given below (you do not need to respond to them). Although I am still not convinced about the use of EC, the manuscript describes the proposed approach clearly.

**We are pleased to note that the discussion led you to reconsider the use of EC, also if you**
**are not fully convinced yet. As we have reported in our answers (AC1) to your comments**
**(RC1), we felt supported in the use of EC by previous published papers stating that not-**
**conservative tracers can be used to create mixing relationship with young water fraction**
**(Kirchner, 2016b) and that results achieved with EC are consistent with those obtained**
**with stable water isotopes (Riazi et al. 2022). Please, see the quote from Kirchner (2016b)**
**and Riazi et al. (2022) that summarize these points with related scientific references:**

*"The young water fraction $F_{yw}$ may also be helpful in inferring chemical processes from*
*streamflow concentrations of reactive chemical species. Because one can determine how $F_{yw}$*
*varies, on average, across different ranges of discharge, one can potentially construct mixing*
*relationships between $F_{yw}$ and the concentrations of reactive species. If the measurable*
*range of Fyw is wide enough, one may even be able to estimate the end-member*
*concentrations corresponding to idealized "young water" ($F_{yw}$ = 1) and "old water" ($F_{yw}$ =*
*0)."*

                                                               *Kirchner (2016b)*

*"EC has been used successfully as a tracer in various previous studies and has compared*
*favourably with results from stable isotopes (Blume et al., 2008; Cano-Paoli et al., 2019;*
*Laudon & Slaymaker, 1997; Meriano et al., 2011; Mosquera et al., 2018). Nevertheless,*
*there are also so characteristics of EC that mean it does not meet the definition of an ideal*
*conservative tracer. One issue is that, as noted above, EC is the net effect of a variety of ions*
*that are influenced by various factors other than age, including geochemical processes*
*within the catchment, leading to some uncertainty regarding its usefulness. For example,*
*ion exchange and weathering likely mean that the ionic composition of water is non-*
*conservative, meaning that EC is also likely to behave non-conservatively (Singha et al.,*
*2011). Nevertheless, taken together, these past studies suggest that EC may provide useful*
*information on water age and hence conditioning travel time model simulations to EC may*
*prove useful."*

                                                                 *Riazi et al. (2022)*

**Accordingly, although EC is not a conservative tracer, it has been used in the past to infer**
**information on water age with successful results. Our results also confirm that, despite**
**the EC limitations (that must be highlighted to use it with care), EC can be used to achieve**
**reliable information on water age.**

**Thanks for pointing out the clarity of our approach description.**

4. You may think about using the list of symbols, because there are many symbols from earlier
works and some other symbols used in your study. Such a list might be helpful to someone
who is not so familiar with all the literature and would like to use your method.

**Thank you for this comment. Yes, there are many symbols in our work and a "List of**
**symbols" could be very useful for the readers: we did not think about it. We will add a**

**"List of symbols" in the revised version of our manuscript if we receive a positive editor response.**

5. It is clear that "old water" in your study is related to the young water fraction (the metric calculated from seasonal isotope variability); i.e., "old water" = 1-young water fraction. However, this term is the same as the "old water" from the isotopic hydrograph separation conducted by a mixing formula. To avoid the confusion, it may be useful to explain, e.g., in the List of symbols that your "old water" is different.

**Thank you for this. Yes, the term "old" is used with different meanings in the scientific literature and this can bring confusion. We will specify what the term "old" means in the revised version of the manuscript. We will do this in the "List of symbols" as you have suggested. We already thought about the use of a different word (e.g., "elderly water fraction = 1 - young water fraction"), but we definitely used the term "old" since it is the term commonly used in past papers about young water fraction.**

6. Despite my comments on the manuscript, if the editor and other reviewer(s) decide that the manuscript can be published, I will not have a problem to accept such a decision.

**We appreciate very much that you have reconsidered your initial decision and that you have provided useful comments that will improve our manuscript.**

7. I agree that you acknowledged many uncertainties related to the use of the method. What I mind is this:

A. We (the hydrological community) know for decades that determination of the input (tracer concentration of the water entering the system, e.g., a catchment) is uncertain. The composition of water infiltrating into the soil that eventually appears in the output (e.g., in catchment runoff) is almost always unknown. We acknowledge this uncertainty and use tracer content in precipitation, because that is what we can (more easily) measure and in sometimes adjust it using different approaches.

B. We know that tracer variability in the input varies both temporally and spatially. The range of temporal variability differs in different years. We acknowledge this uncertainty and approximate the input concentration by the sine curve having the same amplitude over different years. Spatial variability in larger catchments is often neglected.

C. Several approaches are used to estimate the sine curve's amplitude (limiting or accepting the outliers) for weighted or unweighted data. Study periods are sometimes shorter than several years. All this brings the uncertainty which we acknowledge and determine the amplitude.

D. From the amplitudes we calculate the metric (an exact number) characterizing studied system. For many years it was the mean residence/transit time. After the inspiring work by Kirchner (2016) we prefer to use the metric called young water fraction.

E. Young water fraction (an exact number) is defined as "the fraction of runoff with transit times of less than roughly 0.2 years" (Kirchner, 2016). It represents an average over the study period. It seems obvious that when the discharge in a study catchment increases, the young water fraction should likely be greater than in the low flow periods when the streamflow is supplied by water that probably stayed in the catchment longer (we do not know how much longer than 2-3 months, but part of that water may be in the catchment not much longer 2-3 months, i.e. 4, 5, 6?).

F. We introduce a new metric called discharge sensitivity of the young water fraction and assume the exponential relationship between the young water fraction and a virtual young water fraction for discharge equal to zero.

G. It is fascinating and potentially very useful to know how big is the young water fraction on every day, hour, etc. We continue with the development of methodology and calculate daily young water fractions using another, non-conservative tracer (EC) and two-component hydrograph separation. We estimate the unknown tracer concentrations for the two end members though calibration. We assume that there is exponential relationship between the tracer and young water fraction and optimize the daily values so that their average is the same as the young water fraction obtained from seasonal variations of stable isotopes. We acknowledge possible uncertainties.

H. Having the daily young water fractions, we can investigate their relationships with meteorological drivers, and so on and so forth. …..

I. A to H indicate that we are adding uncertainties with every step in the development of our methodology. Please note I am saying "adding" not "accumulating", because I do not know if the uncertainty increases in the described chain of methodology development.

J. We are acknowledging the uncertainty, but continuing to develop the methodology and adding other uncertainties. The result is that since the 1970'/1980' we moved from a simple method providing a rough, but useful characteristic (especially in groundwater hydrology, because it matters if possible pollutant enters an aquifer with mean transit time 6 or 26 months for example) to a complex methodology involving many acknowledged uncertainties providing "exact" numbers for the short time steps.

K. I am not sure how much can the obtained numbers be trusted and whether we are obtaining a substantially new knowledge about the subject of our study, e.g., catchment hydrological cycle (in addition to the information on tracer dynamics). Benetin et al (2022) noted: "In the light of the complexity of the theoretical apparatus underlying time-variant TTDs …, one might wonder if this effort is actually worthwhile and all this complexity is really needed for practical purposes. Our claim is that, while time-variance might not be needed a priori to characterize transport processes in a catchment, it directly affects tracers and solute signals in stream water and plant water. Therefore, acknowledging and incorporating this time variance may be
necessary to capture and explain both high-frequency and long-term tracer dynamics."
**We have understood what you mind. We would like to make some clarification about**
**some points:**
**Data translates to us what nature is saying since we do not speak the language of nature.**
**As with every translation, it is not perfect, but data is the starting point for our research.**
**Sometimes data is sufficient to infer something useful and reliable about how nature**
**works. Some other time, we have to elaborate data by using some methods. Elaborating**
**data (e.g., assuming that input concentration can be represented as a sine curve having**
**the same amplitude over different years) is necessary to extract further information from**
**data or simply to quantify the information that would remain otherwise qualitative.**
**Accordingly, we have to choose the elaboration method that preserves as much as possible**
**the information provided by data. Kirchner (2016a) demonstrated that if we use the**
**isotope data measured in precipitation and streamflow, the convolution approach is not**
**suitable to infer the Mean Transit Time (MTT) as reliable info. MTT is subject to the**
**aggregation error. Thus, Kirchner (2016a) proposed a new metric (the young water**
**fraction) that is not affected by this error (thus, it better preserves the information that**
**measured data can provide us).**
**About hydrologic nonstationarity, Kirchner (2016b) demonstrates that** *"young water*
*fractions can also be estimated separately for individual flow regimes"* **and that** *"one can*
*also estimate the chemical composition of idealized "young water" and "old water" end-*
*members, using relationships between young water fractions and solute concentrations*
*across different flow regimes".*
**Following the statements of Kirchner (2016b), we designed the** *EXPECT* **method.**
**With the** *EXPECT* **method it is possible to estimate the discharge sensitivity of young**
**water fraction differently from the method presented in Gallart et al. (2020). These are**
**two distinct methods with two different uncertainties that can, at the latest, be compared.**
**In this regard, we compared the discharge sensitivity estimated with the** *EXPECT* **method**
**with past estimates of discharge sensitivity (estimated in Gallart et al. 2020) and with**
**flow-specific young water fractions (estimated in von Freyberg et al. 2018).**

- **Our discharge sensitivity estimates are consistent with past discharge sensitivity**
**estimates (estimated in Gallart et al. 2020 with a different method) and with past**
**estimates of flow-specific young water fractions.**
- **The standard errors of the parameters $S^*_d$ and $F^*_0$ are lower than those obtained**
**by using the method of Gallart et al. (2020).**
- **We have obtained additional information on young water and old water EC**
**endmembers.**
- **The mathematical (biunivocal) relationship between $F_{yw}$ and $Q$ of Gallart et al.**
**(2020) does not consider possible hysteretic behavior between discharge and young**

**water fractions during rainfall and after events (Benettin et al. 2017). With the *EXPECT* method we can potentially take into account this behavior by using daily young water fractions estimated from daily EC (that is subject to hysteretic behavior).**

- **We can investigate the short-term variability of young water fractions.**
- **We jointly use stable water isotopes and EC. The latter is not a conservative tracer, but it is measured data that can give information (with some uncertainties) about the water age (Riazi et al. 2022 cum bibl.).**

**From these points we conclude that we are providing a novel method that could potentially provide new insights for new knowledge.**

**Finally, we want to underline that when we talk about water age, we are always dealing with "estimates" since water age cannot be measured. What we can do is to estimate water age based on the use of tracers (stable water isotopes and EC in our study) that can be measured. Indeed, our daily young water fraction estimates cannot be validated by using "water age measurements". However, in the obtained numbers you can trust since we successfully validated our results by using past estimates of flow-specific young water fractions (estimated in von Freyberg et al. 2018) and of discharge sensitivity (estimated in Gallart et al. 2020), both used as a benchmark.**

I have downloaded the discharge and EC data from your catchments and period October 1st, 2010-November 30th, 2015 which is approximately your study period according to Table 1.

1. I agree with you that discharge increase almost always corresponds to EC decrease and vice versa.

2. A few thoughts on the optimized EC values of the endmembers: The low flow periods in the study catchments are never very long (even in winter). Yet, the difference between the optimized EC of the old water fraction in ERL (501 µS.cm-1) and the **minimum (do you mean maximum?)** EC values measured in the stream in period October 2010-November 2015 (334.3 µS.cm-1) is quite high. Even the absolute EC **minimum (do you mean maximum?)** in ERL (439.5 µS.cm-1) between January 1978 and February 2023 (daily data) that was measured on 23rd January 1990, i.e. outside of your study period, was quite different from the optimized value. I am therefore not sure if the optimized EC values are correct. The young water fraction was maybe not very big in January 1990 at catchment discharge of about 0.3 l.s-1. I would assume that streamflow EC would be closer to that of the groundwater, i.e. the measurements over long periods could identify this end member. Similarly, the optimized EC values of the young water fractions seem to be a little higher than data on Central European precipitation suggest (Monteith et al., 2023), but it can be argued that the young water fraction contains some soil water with higher EC.

**This is a key point of our results. You can potentially find the EC of the old water equal to the maximum EC measured in the stream during low flow periods only if the young**

**water fraction is equal to 0 in such flow conditions (i.e., all the streamwater is old water**
**and you can measure the old water endmember), see also Kirchner (2016b):**

*"If the measurable range of $F_{yw}$ is wide enough, one may even be able to estimate the end-*
*member concentrations corresponding to idealized "young water" ($F_{yw} = 1$) and "old water"*
*($F_{yw} = 0$)."*

**From results of flow-specific young water fraction (see Fig. 4 of the preprint), at very low**
**and very high flow conditions the young water fraction resulted to be roughly equal to**
**0.2 and 0.5, respectively, in the three study catchments. This suggests that the streamflow**
**is (very likely) always a mixture of young and old water. Thus, you will never be able to**
**directly measure the old water endmember in the stream. However, the old water EC**
**endmember we have obtained from calibration in the ERL catchment:**

- **is higher than the maximum EC value measured in the stream during the whole**
- **observation period. This makes sense: if streamflow is always a mixture of young**
- **and old water, the old water EC endmember is necessarily higher than the**
- **maximum EC measured in the stream.**
- **is consistent with EC of around 500 μS cm$^{-1}$ in groundwater, measured in the**
- **deepest monitoring well (6.8 m) in the catchment during fall-winter (see lines**
- **246-249 of the preprint)**

**We can do a similar reasoning for young water. Thank you for the reference of data on**
**Central European precipitation. We will include in our discussion that the young water**
**fraction can contain some soil water with higher EC.**

3. According to the coefficient of determination, Q explains about 50% of daily EC variability
in your catchments. It would be great if part of the variability could be explained by young
water fraction. However, how can it be confirmed or rejected if daily young water fractions
were estimated on the basis of EC?

**You can look at median electrical conductivity in specific flow regimes (EC$^Q$) versus flow**
**specific young water fractions (F$^Q_{yw}$) or median discharge in each flow regime (Q$^Q$).**
**Accordingly, EC$^Q$ and F$^Q_{yw}$ have been obtained independently. For example, in the ERL**
**catchment the adjusted R$^2$ obtained by fitting a linear model on EC$^Q$ vs F$^Q_{yw}$ is 0.83, while**
**that obtained by fitting a linear model on EC$^Q$ vs Q$^Q$ is 0.59. This result suggests that the**
**young water fraction explains a larger portion of EC variance than discharges in the ERL**
**catchment.**

References:

Gallart, F., von Freyberg, J., Valiente, M., Kirchner, J. W., Llorens, P., and Latron, J.: Technical note: An improved discharge sensitivity metric for young water fractions, Hydrology and Earth System Sciences, 24, 1101–1107, https://doi.org/10.5194/hess-24-1101-2020, 2020.

Kirchner, J. W.: Aggregation in environmental systems-Part 1: Seasonal tracer cycles quantify young water fractions, but not mean transit times, in spatially heterogeneous catchments, Hydrology and Earth System Sciences, 20, 279–297, https://doi.org/10.5194/hess-20-279-2016, 2016.

Kirchner, J. W.: Aggregation in environmental systems-Part 2: Catchment mean transit times and young water fractions under hydrologic nonstationarity, Hydrology and Earth System Sciences, 20, 299–328, https://doi.org/10.5194/hess-20-299-2016, 2016.

Riazi, Z., Western, A. W., and Bende-Michl, U.: Modelling electrical conductivity variation using a travel time distribution approach in the Duck River catchment, Australia, Hydrological Processes, 36, e14721, https://doi.org/10.1002/hyp.14721, 2022.
von Freyberg, J., Allen, S. T., Seeger, S., Weiler, M., and Kirchner, J. W.: Sensitivity of young water fractions to hydro-climatic forcing and landscape properties across 22 Swiss catchments, Hydrology and Earth System Sciences, 22, 3841–3861, https://doi.org/10.5194/hess-22-3841-2018, 2018.

---

## Author Comment (AC3)

**Dear Anonymous referee #2,**

**Thank you for your care during your reading of the manuscript, your positive remarks**
**and your comments that will help to improve the work. Please, find here below the**
**responses to all your comments.**

**We will take into account all your constructive feedback in the revised version of the**
**manuscript once we receive the editor's response.**

**With kind regards,**

**The Authors**

This article presents an interesting method for estimating the young water fraction based on
high-resolution EC measurements.

**Thanks for the positive overall assessment.**

My only two major concerns are:

1) the authors may consider providing more evidence or referencing literature to support their
three main assumptions for the method.

**Thank you for this comment. In the revised version we will certainly provide more**
**evidence of our assumptions.**

● **The assumption 1 of considering EC as a proxy of water age derives from the**
**following reasoning and literature:**

*"Mineral weathering can be seen as a sequence of complex geochemical reactions*
*driven by properties of fluid flow, such as the contact time between the circulating*
*water and mineral surfaces…"* **(Benettin et al. 2017; Benettin et al. 2015). Thus, the**
**longer the contact time of water with rocks and soils, the higher the mineral**
**weathering. Since EC is a bulk measure of major ions in water, the time that water**
**is retained in a catchment before being released as streamflow (i.e., its age) is**
**expected to be related to the ion concentrations and, accordingly, with EC. Indeed,**
**Mosquera et al. (2016), investigating the mean transit time (MTT) of water and its**
**spatial variability in the wet Andean páramo, found that the mean electrical**
**conductivity is an efficient predictor of mean transit time in this high-elevation**
**tropical ecosystem. Also, Bonacci et al. (2023), analyzing the EC measured in a**
**karst spring, stated that EC can be used to identify the time that water spent in**
**the karst aquifer (Bonacci et al. 2023 cum bibl.). Riazi et al. (2022), modeling the**
**EC variation using a travel time distribution approach, assumed that the salinity**
**of water in catchment storages is a function of water age.**

- **The assumption 2 of considering $EC_{ow}$ higher than $EC_{yw}$ derives from the following reasoning and literature:**

  Following assumption 1, the ion concentrations (i.e., EC) in old (transit times (TT) longer than 2-3 months) water are expected to be higher than the ion concentrations (i.e., EC) in young water with shorter transit times (< 2-3 months). Moreover, young and old streamwater components can derive from different reservoirs in a catchment (Riazi et al. 2022). Among these reservoirs, old water is generally assumed to represent groundwater. This is also supported by the fact that the fraction of baseflow (representing groundwater contribution to streamflow) resulted to be complementary to young water fraction in the framework (including the three Swiss catchments of this study) investigated by Gentile et al. (2023). In this regard, different papers that characterized groundwater EC showed notable differences with EC of precipitation and meltwater. Indeed, Zuecco et al. (2018), by investigating the hydrological processes in an alpine catchment, found that EC of rain water and of recent snow is 19.2 µS/cm and 12.2 µS/cm, respectively. Conversely, they found that groundwater from springs had an EC of 166 µS/cm. Moreover, by investigating the conceptualization of meltwater dynamics in an alpine catchment through hydrograph separation, Penna et al. (2017) defined the snowmelt endmember ranging from 2.9 to 15.3 µS/cm, the glacier melt endmember ranging from 2 to 2.7 µS/cm and the groundwater endmember ranging from 210 to 317.7 µS/cm (average values from springs or streams in fall/winter). These examples are intended to show that groundwater (main source of old water) generally reveals an EC value much higher (around 10-fold) than other sources in a catchment that should preferentially contribute to the young streamwater component. Moreover, Kirchner (2016b) showed the concentrations of reactive chemical species as functions of young water fractions for streams draining three contrasting catchments at Plynlimon, Wales (Fig. 1, extracted from Figure 14 of Kirchner, 2016b and modified after). Calcium concentrations (one of major ions dominating EC in natural streams, Riazi et al., 2022) in streamflow were high for low young water fractions and decreased when young water fractions increased (Fig. 1). By indicating the general trend with gray lines, it is possible to infer the calcium concentration corresponding to $F_{yw} = 0$ (i.e., the old water end-member) which is shown to be higher than theoretical calcium concentration corresponding to $F_{yw} = 1$ (i.e., the young water end-member).

[Figure]

**Fig 1. Calcium concentration as functions of young water fractions for three contrasting catchments at Plynlimon, Wales.**

**Image source: Figure 14 of Kirchner, J. W.: Aggregation in environmental systems-Part 2: Catchment mean transit times and young water fractions under hydrologic nonstationarity, Hydrology and Earth System Sciences, 20, 299–328, https://doi.org/10.5194/hess-20-299-2016, 2016., modified after.**

**Differences in young and old water EC end-members can also be partially justified by looking at differences in event and pre-event water EC endmembers. For example, Cano-Paoli et al. (2019) used the streamwater EC to investigate hydrological processes in alpine headwaters by separating the hydrograph into event and pre-event water. In this regard, they defined the event water end-member equal to 8 µS/cm (as in Penna et al. 2014) and the pre-event water endmember equal to 95 µS/cm (mean value during baseflow conditions). Laudon and Slaymaker (1997), by investigating the hydrograph separation using EC at the lower station of an alpine catchment, defined the rain water EC endmember equal to 6.15 µS/cm and the pre-event water endmember equal to 39 µS/cm. Old (TT > 2-3 months) water is a large fraction of pre-event (TT > few days) water, whereas event water (TT < few days) is a portion of young water (TT < 2-3 months). Due to this overlap (schematized in Fig. 2), would not be surprising a similarity of old and pre-event water EC endmembers, as well as young and event water EC endmembers. However, young and old water EC endmembers are expected to be higher than event and pre-event water EC endmembers, respectively.**

**Fig. 2. Conceptualization of EC variations with streamwater age highlighting the overlap between old and pre-event water, as well as young and event water.**

● **The assumption 3 of using an exponential mixing model that describes how the young water fraction varies with EC in streamwater can be further justified by looking at the relation between flow-specific young water fractions ($F^{Q}_{yw}$) and flow-specific electrical conductivity (see Fig. 1 in the first response to Anonymous referee #1) that we will include in the revised version of the manuscript.**

2) the authors could discuss how their method can be applied to other basins beyond their experimental watersheds.

**Thank you for this comment.**

**We could expand the section 3.3 of the preprint renaming it as ""Limitations of the EXPECT method and recommendations for future applications" in the revised version.**

**We can add a paragraph at the end of section 3.3, briefly outlining the application to other basins beyond our experimental watersheds. We will better explain that a good starting point to choose the mixing model between young water fraction and EC is to visualize the relationship between flow-specific young water fractions and flow-specific electrical conductivities, as suggested in Kirchner (2016b). This relationship could be potentially different from an exponential mixing model. Indeed, the use of the exponential mixing model is not pretended to be the definitive answer to the problem of choosing the right mixing model. However, also if there will be changes in the mathematics, the general method structure to calibrate the endmembers can be applied. Please, note that in some catchments with short and sparse isotope timeseries, flow-specific young water fractions cannot be estimated reliably (von Freyberg et al. 2018).**

I also have some smaller comments as follows:

1. Lines 52-55, readers may seek more detailed descriptions for the terms 'unweighted,' 'flow-weighted,' and 'time-weighted.'

**We have inserted complete information about these terms in the supplementary material, but we missed adding a reference to supplementary material at line 55. We will add this reference. We will also specify that time-weighted or unweighted young water fractions are synonymous). Thank you for this.**

2. Lines 85-86, what do you mean by the 'uncertainty of the discharge sensitivity of the young water fraction'?

**The discharge sensitivity of young water fraction estimation is described in the supplementary material of the preprint. By fitting Eq. (S4) directly to the streamwater isotope values by using the IRLS method it is possible to estimate the parameters $F^*_0$ and $S^*_d$, as well as their associated standard errors. When we talk about the 'uncertainty of the discharge sensitivity of the young water fraction' we are referring to these standard errors. In Table 4 of the preprint we show that with the *EXPECT* method we reduce the standard errors of such parameters.**

3. Table 2, are the numbers of 18O samples and EC samples the same?

**Please, consider that we are not referring to physical samples. We have a daily EC time series obtained from averaging 10-minute data from an EC probe in the stream. When we apply the *EXPECT* method at the "sampling resolution", we subset those EC values from the daily time series that correspond to the time of isotope sampling. In this sense we can say that we have the same number of EC and isotope samples. We will clarify this better in the text. Thank you.**

4. Eqs. 2.1-2.2, I would appreciate more details on the estimation of As, A*s and Ap.

**As for your first minor comment: we have inserted complete information about these terms in the supplementary material, but we missed adding a reference to supplementary material at line 150. We will add this reference.**

5. Figure 5, could you explain what Qmed and Q50/50 represent?

**We will add what $Q_{med}$ and $Q_{50/50}$ represent in the figure caption. Thank you for this.**

6. Figure 6, is the variable snow depth represented as HS in the figure? Please specify the years in each of the panels.

**Yes, $H_S$ and "snow depth" are the same variable. We did not realize that we have used two different names in the figure. We will add "($H_S$)" in the legend after "snow depth". Thank you for having noticed this.**

**We will specify the years in each of the panels.**

7. Figure 7, why not include a scatter plot for Fyw and P, which might better illustrate the correlation?

**The first attempt of Figure 7 was a scatter plot. However, it was not so evident the threshold-like behavior, while it is clear with a binned scatter plot. However, we report hereafter the scatter plot to show how the figure looks like with the representation you suggest:**

[Figure]

8. Line 395, 'significantly reduced the uncertainty of'—how can we observe this reduction in uncertainty from the results section? Please provide more details in the text.

**Following our answer to your second minor comment: in Table 4 of the preprint we show that with the *EXPECT* method we reduce the standard error of the same parameters that can also be obtained with the method presented in Gallart et al. (2020), i.e., the method used to estimate the discharge sensitivity of young water fraction.**

**References:**

**Benettin, P., Bailey, S. W., Rinaldo, A., Likens, G. E., McGuire, K. J., and Botter, G.: Young runoff fractions control streamwater age and solute concentration dynamics, Hydrological Processes, 31, 2982–2986, https://doi.org/10.1002/hyp.11243, 2017.**

**Benettin, P., Bailey, S. W., Campbell, J. L., Green, M. B., Rinaldo, A., Likens, G. E., McGuire, K. J., and Botter, G.: Linking water age and solute dynamics in streamflow at the Hubbard Brook Experimental Forest, NH, USA, Water Resources Research, 51, 9256–9272, https://doi.org/10.1002/2015WR017552, 2015.**

**Bonacci, O. and Roje-Bonacci, T.: Water temperature and electrical conductivity as an indicator of karst aquifer: the case of Jadro Spring (Croatia), Carbonates Evaporites, 38, 55, https://doi.org/10.1007/s13146-023-00881-x, 2023.**

**Cano-Paoli, K., Chiogna, G., and Bellin, A.: Convenient use of electrical conductivity measurements to investigate hydrological processes in Alpine headwaters, Science of The Total Environment, 685, 37–49, https://doi.org/10.1016/j.scitotenv.2019.05.166, 2019.**

**Gentile, A., Canone, D., Ceperley, N., Gisolo, D., Previati, M., Zuecco, G., Schaefli, B.,**
**and Ferraris, S.: Towards a conceptualization of the hydrological processes behind**
**changes of young water fraction with elevation: a focus on mountainous alpine**
**catchments, Hydrology and Earth System Sciences, 27, 2301–2323,**
**https://doi.org/10.5194/hess-27-2301-2023, 2023.**

**Kirchner, J. W.: Aggregation in environmental systems-Part 2: Catchment mean transit**
**times and young water fractions under hydrologic nonstationarity, Hydrology and Earth**
**System Sciences, 20, 299–328, https://doi.org/10.5194/hess-20-299-2016, 2016.**

**Laudon, H. and Slaymaker, O.: Hydrograph separation using stable isotopes, silica and**
**electrical conductivity: an alpine example, Journal of Hydrology, 201, 82–101,**
**https://doi.org/10.1016/S0022-1694(97)00030-9, 1997.**

**Mosquera, G. M., Segura, C., Vaché, K. B., Windhorst, D., Breuer, L., and Crespo, P.:**
**Insights into the water mean transit time in a high-elevation tropical ecosystem,**
**Hydrology and Earth System Sciences, 20, 2987–3004, https://doi.org/10.5194/hess-20-**
**2987-2016, 2016.**

**Penna, D., Engel, M., Bertoldi, G., and Comiti, F.: Towards a tracer-based**
**conceptualization of meltwater dynamics and streamflow response in a glacierized**
**catchment, Hydrology and Earth System Sciences, 21, 23–41,**
**https://doi.org/10.5194/hess-21-23-2017, 2017.**

**Riazi, Z., Western, A. W., and Bende-Michl, U.: Modelling electrical conductivity**
**variation using a travel time distribution approach in the Duck River catchment,**
**Australia, Hydrological Processes, 36, e14721, https://doi.org/10.1002/hyp.14721, 2022.**

**von Freyberg, J., Allen, S. T., Seeger, S., Weiler, M., and Kirchner, J. W.: Sensitivity of**
**young water fractions to hydro-climatic forcing and landscape properties across 22 Swiss**
**catchments, Hydrology and Earth System Sciences, 22, 3841–3861,**
**https://doi.org/10.5194/hess-22-3841-2018, 2018.**

**Zuecco, G., Penna, D., and Borga, M.: Runoff generation in mountain catchments: long-**
**term hydrological monitoring in the Rio Vauz Catchment, Italy, Cuadernos de**
**Investigación Geográfica, 44, 397–428, https://doi.org/10.18172/cig.3327, 2018.**

---

## Author Response (AR1)

**Author's response: a list of all relevant changes made in the manuscript and a point-by-point response to the reviews.**

Alessio Gentile[1], Jana von Freyberg[2,3], Davide Gisolo[1], Davide Canone[1], and Stefano Ferraris[1]

[1]Interuniversity Department of Regional and Urban Studies and Planning (DIST), Politecnico and Università degli Studi di Torino, 10125, Torino, Italy
[2]School of Architecture, Civil and Environmental Engineering, EPFL, 1015, Lausanne, Switzerland
[3]Mountain Hydrology and Mass Movements, Swiss Federal Institute for Forest, Snow and Landscape Research (WSL), 8903, Birmensdorf, Switzerland

*Correspondence to*: Alessio Gentile (alessio.gentile@polito.it )

Dear Editor and Referees,

we would like to thank you for both the overall appreciation of our work and the appreciation of our plan to revise it. Considering the referees' comments, the Editor decided that major revisions are necessary before the review process can be continued. The referees' comments have been very constructive for the paper improvement and served as the guidelines for the changes we made. We have addressed all the issues raised in the interactive discussion including a reorganization and rewriting of some sections to make the text flow more smoothly and to explain our method in a simpler and more effective way.

The present document is subdivided in two Sections. In the first section we summarize all the major changes applied to the submitted document you have revised. In the second section we report a point-by-point response to the reviews.

In the hope of having met your scientific expectations in the revised manuscript, we kindly ask you to reconsider the publication of our work on the Hydrology and Earth System Sciences Journal.

With king regards,

The Authors

**1    List of all relevant changes made in the manuscript.**

**1.1    Abstract**

We have reorganized the abstract to effectively summarize our method. In particular:

-    we have highlighted the main assumptions through a bullet-point list.

- we have emphasized at line 33 that the hydrograph separation in our method is 'unconventional' to avoid confusion with the traditional separation into event and pre-event water.
- From line 40 to 45, we have summarized the results, emphasizing that we validated the values of the endmembers obtained from calibration.
- From line 46 to 48, we have underscored that the manuscript outlines the main limitations of the method along with recommendations for its application in catchments different from those investigated in this study.

**1.2   Introduction**

From line 50 to line 100, some parts have been simply rewritten to make the text more fluent or clearer. Also, Eq. (1) has been further detailed to clarify the method of Gallart et al. (2020) for readers who may not have read the relative paper. From line 89 to line 137, the introduction has been extensively revised, integrating new information requested by the Editor and reviewers:

- From line 89 to line 100, the advantages and limitations of both EC and isotopes are explained, showing that they have complementary characteristics and could be used together for various applications.
- From line 101 to line 114, we have presented several articles from the scientific literature suggesting the use of electrical conductivity as a proxy for water age (thus giving support to our assumption of using EC as a proxy of the water age for a time-source hydrograph separation)
- Accordingly, from line 115 to line 134, we have included many articles (from 1997 to 2023) from the scientific literature employing EC for time-source hydrograph separation and showing good agreement with results obtained using stable water isotopes as requested by the editor and referees.

**1.3   Material and methods**

**1.3.1   Study sites and data set**

This section has remained almost unchanged. From line 179 to line 186, we have provided a more detailed explanation of how flow-specific young water fractions can be estimated for readers who may not have read the articles by Kirchner et al. (2016) and von Freyberg et al. (2018). We then changed Figure 2 so that the colors of the three basins are consistent with those shown in the new Figure 3. In Table 2, we added a column which indicates the median electrical conductivity in each flow regime for the three studied catchments.

**1.3.2   The *EXPECT* method: two-component *E*lectrical *C*onductivity-based hydrograph separa*T*ion employing an *EXP*onential mixing model**

This section has been extensively reorganized and rewritten to explain the method more simply and fluently (as requested by the Editor). Indeed, some paragraphs have been rearranged to help the reader follow the logical thread underlying our method. Moreover, we added an analysis showing how flow-specific young water fractions vary with the median flowspecific EC (reported in Table 2), illustrated in the new Figure 3 of the revised manuscript. This analysis serves both to justify the choice of an exponential mixing model, providing further support (as requested by both reviewers) for our hypotheses, and to provide an approximate estimate of the endmembers ($EC_{ow}^{raw}, EC_{yw}^{raw}$), which will be compared with those obtained from the calibration procedure. This analysis also demonstrates why choosing a linear mixing model would not be suitable for the three basins under study.

By reorganizing the paragraphs, some equations have been moved earlier, and thus, the equation numbers have been updated both in the text and in Fig. 4 of the revised manuscript.

**1.4 Results and Discussion**

**1.4.1 Physical likelihood of calibrated endmembers and discharge sensitivity of young water fraction**

This section has also been extensively reorganized since we have integrated many pieces of information requested by the Editor and reviewers regarding the validation of the obtained endmembers. From line 320 to line 348, we have included several published works supporting the difference of several orders of magnitude between the electrical conductivity of old water ($EC_{ow}$) and that of young water ($EC_{yw}$). These studies support our results and our initial hypothesis of considering $EC_{yw} < EC_{ow}$. We have always indicated the types of basins studied in the cited articles, which in most cases are alpine basins. From line 353 to line 357, we have pointed out how the value of the calibrated endmembers is consistent with the value of the endmembers obtained from the analysis illustrated in Figure 3. We have also discussed what discrepancies may be due to, also integrating the observations made by anonymous reviewer #1 (lines 357-359). From line 365 to line 375, we have supported the fact that the calibrated endmembers are higher and lower than the maximum and minimum EC measured in the stream, respectively, is reasonable. Moreover, we added three columns in Table 3 reporting the catchment ID, $EC_{yw}^{raw} \pm SE$, $EC_{ow}^{raw} \pm SE$, where SE indicate the standard error. This allows comparing the endmember values obtained from calibration with those obtained from the analysis shown in Figure 3. The latter have also been reported in Figure 5 along with the measured electrical conductivity values in two wells, one inside ERL and the other nearby.

**1.4.2 An immediate application of the EXPECT method: flow duration curves of young/old water and the temporal variability of young water fractions.**

This section has remained almost unchanged. We have simply explained the meaning of $Q_{50/50}$ in the label of Figure 6 and some little modification to Figure 7 as requested by the anonymous referee #2.

**1.4.3 Limitations of the EXPECT method**

This section has been expanded. Firstly, we changed its title from 'Limitations of the EXPECT method' to 'Limitations of the EXPECT and recommendations for future applications.' As requested by reviewer #2, we explained (from line 471 to line 490) how it is possible to apply and/or adapt the methodology presented in this work to different basins, emphasizing the

precautions that need to be taken into account. At lines 496-497, we highlighted the importance of estimating the uncertainty of the endmembers and, consequently, of the fractions of young water that will be estimated with these endmembers.

**1.5    Summary and Conclusions**

This section has been partially rewritten to highlight the main aim and findings of this work.

**1.6    Appendix A**

This section has remained almost unchanged.

**1.7    List of symbols**

We have added the list of symbols as recommended by reviewer #2. This list should help the reader not to get lost with the symbols presented in the manuscript and clarify the meaning of the terms used.

**2    Response to Referees**

**2.1    Response to referee #1**

Dear authors,

I would like to thank you for the effort put into addressing my comments. The discussion is the best way to clarify the ideas and realize possible misunderstandings or drawbacks. Our discussion can be perhaps useful also to journal readers.

**Dear Anonymous referee #1,**

**We thank you very much for your reply to our comments (AC1, https://doi.org/10.5194/egusphere-2023-1797-AC1) that further stimulates the discussion. We are pleased to note that the discussion has solved some possible misunderstandings and brought constructive comments and feedback to our manuscript. Accordingly, we have incorporated all your constructive feedback in the revised version of the manuscript that have contributed significantly to improving the work.**

**Please, find below a point-by-point response to your comments.**

**Sincerely,**

**The Authors**

1. I understood that you did not do hydrograph separation with stable isotopes. It is not necessary to rewrite lines 165-172. The reader can obtain more detailed information from your response to my comments.

125 **Ok, thank you for this.**

2. The key assumption of your approach is the exponential relationship between EC and young water fraction. Could you try to justify it also in some other way than just mathematically (l. 176-190)?

130 **Thank you for this comment. We have realized that the exponential relationship between EC and young water fraction could not appear robustly justified as presented in the preprint. In this regard, we have added a new analysis showing how the median flow-specific EC varies along with flow-specific young water fractions ($F^Q_{yw}$). This analysis is reported in Figure 3 of the revised manuscript. From this figure it is possible to visualize the relationship between electrical conductivity and young water fraction. Accordingly, from line 213 to line 216 we have written: "As visible**

135 **in Fig. 3, the relationship between $F^Q_{yw}$ and median flow-specific EC is well described by an exponential mixing model. Indeed, the widely used linear mixing model proves to be poorly suited here since it is pointing to a negative EC endmember of young water (i.e., EC value corresponding to $F^Q_{yw}= 1$, Fig. 3). This will be thoroughly discussed in the Appendix A."**

140 3. I have downloaded and checked the discharge and EC data for your catchments. Some thoughts are given below (you do not need to respond to them). Although I am still not convinced about the use of EC, the manuscript describes the proposed approach clearly.

**We are pleased to note that the discussion led you to reconsider the use of EC, also if you are not fully convinced yet.**
145 **We are supported in the use of EC by:**

- **Kirchner (2016b) statement about the use of not-conservative tracers to create mixing relationship with young water fraction. Please, see the quote from Kirchner (2016b):**
*"The young water fraction $F_{yw}$ may also be helpful in inferring chemical processes from streamflow concentrations of reactive chemical species. Because one can determine how $F_{yw}$ varies, on average, across*
150 *different ranges of discharge, one can potentially construct mixing relationships between $F_{yw}$ and the* *concentrations of reactive species.* *If the measurable range of $F_{yw}$ is wide enough, one may even be able to estimate the end-member concentrations corresponding to idealized "young water" ($F_{yw}$ = 1) and "old water" ($F_{yw}$ = 0)."*
**We have reported this from line 201 to line 204 of the revised manuscript.**

155

- **EC provided useful information on water age in past studies and EC-based hydrograph separation results were favorably compared with those obtained with isotope-based hydrograph separation.**

    **In this regard, we have included from line 101 to line 134 many published papers supporting the use of EC.**

160 **Thanks for pointing out the clarity of our approach description.**

4. You may think about using the list of symbols, because there are many symbols from earlier works and some other symbols used in your study. Such a list might be helpful to someone who is not so familiar with all the literature and would like to use your method.

165

**Thank you for this comment. We agree that there are many symbols in our work and a "List of symbols" is very useful for the readers. In this regard, we have added a "List of symbols" in the revised version of our manuscript. Please, see lines from 572 to 664.**

170 5. It is clear that "old water" in your study is related to the young water fraction (the metric calculated from seasonal isotope variability); i.e., "old water" = 1-young water fraction. However, this term is the same as the "old water" from the isotopic hydrograph separation conducted by a mixing formula. To avoid the confusion, it may be useful to explain, e.g., in the List of symbols that your "old water" is different.

175 **Thank you for this. Yes, the term "old" is used with different meanings in the scientific literature and this can bring confusion. We have specified in the text that the term "old" means "with transit times higher than 2-3 months" (e.g., lines 30-31, lines 228-229). Please, see also the definition of "old water" given at line 634 of the List of Symbols reported in the revised version of the manuscript.**

180 6. Despite my comments on the manuscript, if the editor and other reviewer(s) decide that the manuscript can be published, I will not have a problem to accept such a decision.

**We appreciate very much that you have reconsidered your initial decision and that you have provided useful comments that improved our manuscript. Considering the major changes applied to the revised version of the**
185 **manuscript following your comments, we hope to have met the scientific expectations required for publication in HESS.**

7. I agree that you acknowledged many uncertainties related to the use of the method. What I mind is this:

190

A. We (the hydrological community) know for decades that determination of the input (tracer concentration of the water entering the system, e.g., a catchment) is uncertain. The composition of water infiltrating into the soil that eventually appears in the output (e.g., in catchment runoff) is almost always unknown. We acknowledge this uncertainty and use tracer content in precipitation, because that is what we can (more easily) measure and in sometimes adjust it using different approaches.

195

B. We know that tracer variability in the input varies both temporally and spatially. The range of temporal variability differs in different years. We acknowledge this uncertainty and approximate the input concentration by the sine curve having the same amplitude over different years. Spatial variability in larger catchments is often neglected.

200    C. Several approaches are used to estimate the sine curve's amplitude (limiting or accepting the outliers) for weighted or unweighted data. Study periods are sometimes shorter than several years. All this brings the uncertainty which we acknowledge and determine the amplitude.

D. From the amplitudes we calculate the metric (an exact number) characterizing studied system. For many years it was the
205    mean residence/transit time. After the inspiring work by Kirchner (2016) we prefer to use the metric called young water fraction.

E. Young water fraction (an exact number) is defined as "the fraction of runoff with transit times of less than roughly 0.2 years" (Kirchner, 2016). It represents an average over the study period. It seems obvious that when the discharge in a study
210    catchment increases, the young water fraction should likely be greater than in the low flow periods when the streamflow is supplied by water that probably stayed in the catchment longer (we do not know how much longer than 2-3 months, but part of that water may be in the catchment not much longer 2-3 months, i.e. 4, 5, 6?).

F. We introduce a new metric called discharge sensitivity of the young water fraction and assume the exponential
215    relationship between the young water fraction and a virtual young water fraction for discharge equal to zero.

G. It is fascinating and potentially very useful to know how big is the young water fraction on every day, hour, etc. We continue with the development of methodology and calculate daily young water fractions using another, non-conservative tracer (EC) and two-component hydrograph separation. We estimate the unknown tracer concentrations for the two end
220    members though calibration. We assume that there is exponential relationship between the tracer and young water fraction

and optimize the daily values so that their average is the same as the young water fraction obtained from seasonal variations of stable isotopes. We acknowledge possible uncertainties.

H. Having the daily young water fractions, we can investigate their relationships with meteorological drivers, and so on and so forth. …..

I. A to H indicate that we are adding uncertainties with every step in the development of our methodology. Please note I am saying "adding" not "accumulating", because I do not know if the uncertainty increases in the described chain of methodology development.

J. We are acknowledging the uncertainty, but continuing to develop the methodology and adding other uncertainties. The result is that since the 1970'/1980' we moved from a simple method providing a rough, but useful characteristic (especially in groundwater hydrology, because it matters if possible pollutant enters an aquifer with mean transit time 6 or 26 months for example) to a complex methodology involving many acknowledged uncertainties providing "exact" numbers for the short time steps.

K. I am not sure how much can the obtained numbers be trusted and whether we are obtaining a substantially new knowledge about the subject of our study, e.g., catchment hydrological cycle (in addition to the information on tracer dynamics). Benetin et al (2022) noted: "In the light of the complexity of the theoretical apparatus underlying time-variant TTDs …, one might wonder if this effort is actually worthwhile and all this complexity is really needed for practical purposes. Our claim is that, while time-variance might not be needed a priori to characterize transport processes in a catchment, it directly affects tracers and solute signals in stream water and plant water. Therefore, acknowledging and incorporating this time variance may be necessary to capture and explain both high-frequency and long-term tracer dynamics."

**We have understood what you mind. We would like to make some clarification about some points:**

**We recognize the challenges in determining the input tracer concentration and the temporal and spatial variability (Point A and B) of tracer content in the input, which is often neglected (Point B). However, the data uncertainty remains regardless of the method we use to process them. Accordingly, we have to choose the elaboration method that preserves as much as possible the information provided by data. Kirchner (2016a) demonstrated that if we use the isotope data measured in precipitation and streamflow, the convolution approach is not suitable to infer the Mean Transit Time (MTT) as reliable info (Point J), since it is subject to the aggregation error. Thus, Kirchner (2016a) proposed a new metric, the young water fraction, that is not affected by this error. Following Kirchner (2016a), the**

young water fraction, and not MTT, is the information we can reliably extract from seasonal tracer cycles. Indeed, also 6 or 26 months to which you are referring are exact numbers with an uncertainty that, according to Kirchner (2016a), is much higher than those we can obtain from estimating the young water fraction from the amplitude ratio approach. Nevertheless, we agree that by neglecting the temporal variability of tracer input, e.g., assuming that input concentration can be represented as a sine curve having the same amplitude over different years, is a simplifying assumption, but it is a starting point to estimate quantities more reliable than MTT.

Following the key works of Kirchner (2016a, 2016b), the young water fraction has become a cornerstone, and the methodological chain has continued from this point. Accordingly, following the paper of Kirchner (2016b), the concept of discharge sensitivity of young water fraction has been developed by von Freyberg et al. (2018) and improved by Gallart et al. (2020b). Similarly, starting from Kirchner's paper (2016b), we have developed our own methodology that also allows for the estimation of discharge sensitivity. These are two distinct methods with two different uncertainties that can, at the latest, be compared.

In the revised version, we have validated our results about the optimized endmembers and the daily/sampling young water fraction of which we compute the uncertainty. Please see section 3.1 and Fig. 5.  As reported at the point 2, in the revised version we have added a new analysis showing how the median flow-specific EC varies along with flow-specific young water fractions. This analysis allows us to have a first-order estimate of the endmembers (see lines from 216 to 221 of the revised version) that have been used as a benchmark compared to those calibrated. Moreover, we have validated the daily/sampling young water fractions (white-brown points in Fig. 5) with both flow-specific young water factions and the exponential fit with parameters previously obtained by Gallart et al. (2020b) (see black solid line in Fig. 5) that we use as benchmark. Our results favourably compared with the considered benchmarks. Accordingly, we retain that our results are reliable, and you can trust in the obtained quantities.

I have downloaded the discharge and EC data from your catchments and period October 1st, 2010-November 30th, 2015 which is approximately your study period according to Table 1.

1. I agree with you that discharge increase almost always corresponds to EC decrease and vice versa.

2. A few thoughts on the optimized EC values of the endmembers: The low flow periods in the study catchments are never very long (even in winter). Yet, the difference between the optimized EC of the old water fraction in ERL (501 µS.cm-1) and the **minimum (do you mean maximum?)** EC values measured in the stream in period October 2010-November 2015 (334.3 µS.cm-1) is quite high. Even the absolute EC **minimum (do you mean maximum?)**  in ERL (439.5 µS.cm-1) between January 1978 and February 2023 (daily data) that was measured on 23rd January 1990, i.e. outside of your study period, was quite different from the optimized value. I am therefore not sure if the optimized EC values are correct. The young water

fraction was maybe not very big in January 1990 at catchment discharge of about 0.3 l.s-1. I would assume that streamflow
290  EC would be closer to that of the groundwater, i.e. the measurements over long periods could identify this end member. Similarly, the optimized EC values of the young water fractions seem to be a little higher than data on Central European precipitation suggest (Monteith et al., 2023), but it can be argued that the young water fraction contains some soil water with higher EC.

295  **Thank you for this comment since this is a key point of our results. You can potentially find the EC of the old water equal to the maximum EC measured in the stream during low-flow periods only if the young water fraction is equal to 0 in such flow conditions (i.e., all the streamwater is old water and you can directly measure in the stream the old water endmember). This is not the case of our three study catchments. We report here what we have written from lines to 365 to 375 of the revised manuscript:**

300  **"Our method estimates the EC endmember values for the cases $F_{yw}(t_i) = 1$ and $F_{yw}(t_i) = 0$ that are generally difficult to determine experimentally, thus providing additional information about young and old water in the systems under study. In this regard, in each one of the three study sites, the theoretical endmembers $EC_{yw}^{opt}$ are lower than the minimum EC value measured in the streams; analogously, the calibrated $EC_{ow}^{opt}$ values are higher than the maximum measured EC value (boxplots *versus* horizontal dashed lines in Fig. 5). This is expected for a natural,**
305  **heterogeneous system where incoming precipitation mixes with stored water, and thus streamwater never contains 100% young or old water, respectively. Instead, streamwater is a mixture of these two components. This is supported by the fact that $F^{Q}_{yw}$ cover only a limited range of young water fractions (roughly from 0.1 to 0.5). This result demonstrates that the choice of the old water endmember based on tracer values sampled during baseflow conditions can result in an underestimation of the theoretical old water endmember. Although these stream conditions suggest**
310  **the prevalence of old water, if the percentage of old water is less than 100%, then the measured tracers still reflect some mixing (albeit limited) with young water."**

**In the revised manuscript we have included your comments about the fact that the optimized EC values of the young water are a little higher than data on Central European precipitation (Monteith et al., 2023), and that this can be**
315  **explained by considering the presence of soil water with higher EC. Please see lines 357-359 of the revised manuscript. Thank you for this.**

3. According to the coefficient of determination, Q explains about 50% of daily EC variability in your catchments. It would be great if part of the variability could be explained by young water fraction. However, how can it be confirmed or rejected if
320  daily young water fractions were estimated on the basis of EC?

You can look at median electrical conductivity in specific flow regimes versus flow specific young water fractions ($F^Q_{yw}$) or median discharge in each flow regime. Accordingly, electrical conductivity in specific flow regimes and $F^Q_{yw}$ have been obtained independently. For example, in the ERL catchment the adjusted $R^2$ obtained by fitting a linear model on electrical conductivity in specific flow regimes vs $F^Q_{yw}$ is 0.83, while that obtained by fitting a linear model on electrical conductivity in specific flow regimes vs median discharge in each flow regime is 0.59. This result suggests that the young water fraction explains a larger portion of EC variance than discharges in the ERL catchment.

**2.2 Response to referee #2**

**Dear Anonymous referee #2,**

**Thank you for your care during your reading of the manuscript, your positive remarks and your comments that helped to improve the work a lot. We have implemented all your constructive feedback in the revised version of the manuscript. Please, find here below the responses to all your comments.**

**With kind regards,**

**The Authors**

This article presents an interesting method for estimating the young water fraction based on high-resolution EC measurements.

**Thanks for the positive overall assessment.**

My only two major concerns are:

1) the authors may consider providing more evidence or referencing literature to support their three main assumptions for the method.

**Thank you for this comment. In the revised version we have provided more evidence of our assumptions.**

- **The assumption of considering an exponential mixing model for hydrograph separation has been robustly justified in the revised manuscript. Please, see the analysis reported in Fig. 3 of Section 2.2 and the Appendix A of the revised manuscript. We report here what we have written in lines from 213 to 216: "As visible in Fig. 3, the relationship between $F^Q_{yw}$ and median flow-specific EC is well described by an exponential mixing model. Indeed, the widely used linear mixing model proves to be poorly suited here since it is pointing to a**

355           negative EC endmember of young water (i.e., EC value corresponding to $F^Q_{yw}= 1$, Fig. 3). This will be thoroughly discussed in the Appendix A."

- The assumption of considering EC as a proxy of water age for a time-source hydrograph separation has been widely supported by past papers we have included in the revised version of the manuscript. Please, see lines from 101 to 134 that I report hereafter:

[revised manuscript text omitted]

Moreover, Kirchner (2016b) showed the concentrations of reactive chemical species as functions of young water fractions for streams draining three contrasting catchments at Plynlimon, Wales (Fig. 1, extracted from Figure 14 of Kirchner, 2016b and modified after). Calcium concentrations (one of major ions dominating EC in natural streams, Riazi et al., 2022) in streamflow were high for low young water fractions and decreased when young water fractions increased (Fig. 1). By indicating the general trend with gray lines, it is possible to infer the calcium concentration corresponding to $F_{yw} = 0$ (i.e., the old water end-member) which is shown to be higher than theoretical calcium concentration corresponding to $F_{yw} = 1$ (i.e., the young water end-member).

[Figure]

**Fig 1. Calcium concentration as functions of young water fractions for three contrasting catchments at Plynlimon, Wales.**

**Image source: Figure 14 of Kirchner, J. W.: Aggregation in environmental systems-Part 2: Catchment mean transit times and young water fractions under hydrologic nonstationarity, Hydrology and Earth System Sciences, 20, 299–328, https://doi.org/10.5194/hess-20-299-2016, 2016., modified after.**

2) the authors could discuss how their method can be applied to other basins beyond their experimental watersheds.

**Thank you for this comment.**

**In the revised version, we expanded the section 3.3 of the preprint and we have renamed it as ""Limitations of the EXPECT method and recommendations for future applications". Please, see lines from 470 to 490 we report hereafter:**

**"While the *EXPECT* method can offer valuable insights into the young water fraction's discharge sensitivity and its time-variability, it is not without its limitations. The assumption of considering EC as a proxy of streamwater age may not hold true in all hydrological systems. For example, human activities, such as mining, irrigation or wastewater inputs can alter the streamwater EC in unpredictable ways. Another example involves catchments with highly soluble rocks in the aquifers (e.g., limestone or gypsum), that are susceptible to dissolution by water. It has been shown that EC can increase with $Q$ in some karst systems due to remobilization of the circulating water in the fractured areas (Balestra et al., 2022). Therefore, the $F_{yw}$-EC relationship (Eq. 3) can be very different from that in our three study catchments that are mainly groundwater influenced. Indeed, also an early study advised to be mindful of EC behaviour since it depends on specific characteristics of each catchment (Laudon and Slaymaker, 1997). Accordingly, for future applications of the method presented in this paper, we**

**recommend to start visualizing the relationship between flow-specific young water fractions and flow-specific electrical conductivities with the aim of constructing a site-specific mixing relationship, as suggested by Kirchner (2016b). Please, note that this relationship could be potentially different from an exponential mixing model. Indeed, the use of the exponential mixing model is not pretended to be the definitive answer to the problem of**

475 **choosing the right mixing model for flow partitioning in young and old water. Accordingly, if the most suitable mixing model turns out to be different from an exponential mixing model, the equations presented in this study will need to be adapted to the specific case study. However, the method's application scheme for calibrating the endmembers can still be employed. Nevertheless, in some catchments with short and sparse isotope timeseries, flow-specific young water fractions cannot be estimated reliably (von Freyberg et al., 2018b). von Freyberg et al.**

480 **(2018a) were able to estimate reliable flow-specific young water fractions for nine Swiss catchments that disposed of isotope timeseries 4 to 5 years-long with a minimum number of samples from 81 to a maximum of 140, where streamwater grab samples were collected approximately fortnightly. Thus, we suggest an isotope data set with these characteristics to construct a reliable site-specific mixing model with both flow-specific EC and $F^{Q}_{yw}$.”**

485 1. Lines 52-55, readers may seek more detailed descriptions for the terms 'unweighted,' 'flow-weighted,' and 'time-weighted.'

**We have inserted complete information about these terms in the supplementary material and we add the reference to supplementary material at line 67. In order to avoid confusion, we have only used the term "unweighted" in the revised version. Thank you for this.**

490

2. Lines 85-86, what do you mean by the 'uncertainty of the discharge sensitivity of the young water fraction'?

**The estimation of the discharge sensitivity of young water fraction is described in the supplementary material and (briefly) in lines from 69 to 82. Referring to the supplementary material, by fitting Eq. (S4) directly to the streamwater isotope values by using the IRLS method it is possible to estimate the parameters**

495 **$F^{*}_{0}$ and $S^{*}_{a}$, as well as their associated standard errors. When we talk about the 'uncertainty of the discharge sensitivity of the young water fraction' we are referring to these standard errors. In Table 4 of the revised version we show that with the EXPECT method we reduce the standard errors of such parameters.**

3. Table 2, are the numbers of $^{18}O$ samples and EC samples the same?

500 **We have explained this at lines 293-298 of the revised version:**

**"At SR, please note that the "EC samples" are not referring to physical samples in this specific application. Accordingly, $EC(t_i)$ and $Q(t_i)$ are obtained by sub-setting those EC and $Q$ values from the daily time series that correspond to the time of isotope sampling. In this sense, we can say that the number of EC samples and isotope samples is the same. Nevertheless, the method can be potentially applied at SR in catchments in**

505 **which EC is only measured from water samples. At SR, $F_{yw}(t_i)$ values are estimated only for those days on which an isotope sample was taken."**

4.  Eqs. 2.1-2.2, I would appreciate more details on the estimation of As, A*s and Ap.

    **As for your first minor comment: we have inserted complete information about these terms in the**
510 **supplementary material and we add the reference to supplementary material at line 67.**

5.  Figure 5, could you explain what Qmed and Q50/50 represent?

    **We have added what $Q_{med}$ and $Q_{50/50}$ represent in the figure caption. Thank you for this.**

515 6.  Figure 6, is the variable snow depth represented as HS in the figure? Please specify the years in each of the panels.

    **Yes, HS and "snow depth" are the same variable. We did not realize that we have used two different names in the figure. We have added "(HS)" in the legend after "snow depth" and we have specified the years in each of the panels. Thank you for your comment.**

520 7.  Figure 7, why not include a scatter plot for $F_{yw}$ and P, which might better illustrate the correlation?

    **The first attempt of Figure 7 was a scatter plot. However, it was not so evident the threshold-like behavior, while it is clear with a binned scatter plot. However, we report hereafter the scatter plot to show how the figure looks like with the representation you suggest:**

[Figure]

525

8. Line 395, 'significantly reduced the uncertainty of'—how can we observe this reduction in uncertainty from the results section? Please provide more details in the text.

530

> **Following our answer to your second minor comment: in Table 4 of the revised version, we show that with the *EXPECT* method we reduce the standard error of the same parameters that can also be obtained with the method presented in Gallart et al. (2020), i.e., the existing method used to estimate the discharge sensitivity of young water fraction.**

535